# The *CIC-ERF* co-deletion underlies fusion-independent activation of ETS family member, ETV1, to drive prostate cancer progression

**Nehal Gupta[1], Hanbing Song[1], Wei Wu[1], Rovingaile K Ponce[1], Yone K Lin[1], Ji Won Kim[1], Eric J Small[1,2], Felix Y Feng[1,2,3], Franklin W Huang[1,2]\*, Ross A Okimoto[1,2]\***

[1]Department of Medicine, University of California, San Francisco, United States; [2]Helen Diller Family Comprehensive Cancer Center, University of California, San Francisco, United States; [3]Department of Radiation Oncology, University of California, San Francisco, United States

**Abstract** Human prostate cancer can result from chromosomal rearrangements that lead to aberrant ETS gene expression. The mechanisms that lead to fusion-independent ETS factor upregulation and prostate oncogenesis remain relatively unknown. Here, we show that two neighboring transcription factors, Capicua (*CIC*) and ETS2 repressor factor (*ERF*), which are co-deleted in human prostate tumors can drive prostate oncogenesis. Concurrent *CIC* and *ERF* loss commonly occur through focal genomic deletions at chromosome 19q13.2. Mechanistically, *CIC* and *ERF* co-bind the proximal regulatory element and mutually repress the ETS transcription factor, *ETV1*. Targeting ETV1 in *CIC* and *ERF*-deficient prostate cancer limits tumor growth. Thus, we have uncovered a fusion-independent mode of ETS transcriptional activation defined by concurrent loss of *CIC* and *ERF*.

**\*For correspondence:**
franklin.huang@ucsf.edu (FWH);
ross.okimoto@ucsf.edu (RAO)

**Competing interest:** The authors declare that no competing interests exist.

## Editor's evaluation

This study provides insight into a potentially new genetically defined subset of prostate tumors driven by concurrent loss of the ERF and CIC tumor suppressor genes, in the absence of the canonical fusion event involving TMPRSS2 (around 10% of all cases). The work both validates previous findings and provides new data that support a compelling overall conclusion that combined ERF and CIC loss promotes prostate tumorigenesis by increasing expression of the oncogenic driver ETV1. This is an important study based on convincing evidence, that will be of interest to researchers in the field of prostate cancer.

## Introduction

Prostate cancer (PCa) is the most common solid tumor malignancy in men. Activation of ETS transcription factors (TFs), *ERG*, *ETV1*, *ETV4*, and *ETV5*, are present in approximately 60% of PCa, underscoring their importance in prostate oncogenesis (*Nelson et al., 2003*). In human PCa, ETS TFs are most commonly activated through gene rearrangements that fuse the androgen responsive gene, *TMPRSS2*, to either *ERG*, *ETV1*, *ETV4*, or *ETV5* (*Sizemore et al., 2017*). Beyond ETS TF fusions, little is known about the underlying molecular mechanisms that lead to increased expression of wildtype (WT) ETS factors, which confer aggressive malignant phenotypes and associate with poor clinical outcomes in fusion negative PCa patients (*Baena et al., 2013*).

Capicua (CIC) is a High-mobility group (HMG) box TF that silences *ETV1*, *ETV4*, and *ETV5* through direct target gene repression (*Kim et al., 2021*; *Simón-Carrasco et al., 2018*). *CIC* is frequently altered in human cancer, where it functionally suppresses tumor growth and metastasis (*Ahmad et al., 2019*; *Bettegowda et al., 2011*; *Choi et al., 2015*; *Dissanayake et al., 2011*; *Kim et al., 2018*; *Okimoto et al., 2017*; *Seim et al., 2017*; *Yoshiya et al., 2021*). Notably, in PCa, *CIC* is commonly altered through genomic loss (homozygous and heterozygous deletion) in ~10% of PCa patients (*Abida et al., 2019*; *Grasso et al., 2012*; *Hieronymus et al., 2014*; *Huang et al., 2017*; *Robinson et al., 2015*; *Cancer Genome Atlas Research Network, 2015*) and inactivation of *CIC* de-represses *ETV1*, *ETV4*, and *ETV5* transcription to promote tumor progression (*Bettegowda et al., 2011*; *Choi et al., 2015*; *Kim et al., 2018*; *Okimoto et al., 2017*). Leveraging mutational profiling data from multiple PCa cohorts, we previously observed concurrent loss of the ETS2 repressor factor (*ERF*) in *CIC*-deficient prostate tumors (*Huang et al., 2017*). Combinatorial loss is most commonly the result of focal deletions (homozygous and heterozygous) at the 19q13.2 locus, where *CIC* and *ERF* are physically adjacent (long and short isoforms of *CIC* are separated from *ERF* by approximately 15 and 30 kb, respectively) to one another in the genome. Since ERF is a transcriptional repressor that binds ETS DNA motifs (*Bose et al., 2017*), we hypothesized that in a fusion independent manner, CIC and ERF cooperate to mutually suppress *ETS* target genes in PCa.

Through an integrative genomic and functional analysis, we mechanistically show that CIC and ERF directly bind and co-repress a proximal *ETV1* regulatory element limiting PCa progression. Concomitant loss of CIC and ERF de-represses *ETV1*-mediated transcriptional programs and confer *ETV1* dependence in multiple PCa model systems. Thus, we reveal a fusion-independent mechanism to de-repress ETS-mediated PCa progression and potentially uncover a therapeutic approach to target *CIC-ERF* co-deleted PCa.

## Results

CIC is a TF that directly suppresses *ETV1*, *ETV4*, and *ETV5* TF family members (*Futran et al., 2015*; *Jiménez et al., 2012*; *Kim et al., 2021*; *Okimoto et al., 2017*). CIC silences target genes through direct binding of a highly conserved DNA-binding motif (T[G/C]AAT[G/A]AA; *Figure 1A*; *Ajuria et al., 2011*; *Futran et al., 2015*; *Jiménez et al., 2012*). *CIC* is commonly altered in multiple human cancer subtypes where it suppresses tumor growth and metastasis (*Kim et al., 2021*). *CIC* is located on chromosome 19q13.2, directly adjacent to another transcriptional repressor, namely the *ERF* (*Figure 1B*). ERF binds and competes for ETS TF-binding sites (GGAA-motifs) and is frequently altered in human PCa, predominantly through focal deletions (*Bose et al., 2017*; *Hou et al., 2020*). We thus hypothesized that concurrent loss of *CIC* and *ERF* may de-repress an ETS-driven transcriptional program that drives PCa progression in a fusion-independent manner. To explore this, we first queried 15 PCa datasets curated on cBioPortal (*Cerami et al., 2012*; *Gao et al., 2013*) and identified a high cooccurrence rate (p<0.001, two-sided Fisher's exact test [FET]) for *CIC* (10%) and *ERF* (12%) homozygous and heterozygous deletions (*Figure 1C*), suggesting that concurrent loss occurs through focal copy number change at the 19q13.2 locus. Through analysis of these clinically annotated specimens, we observed that the *CIC-ERF* co-deletion was present at an increased frequency in PCa with higher Gleason scores and later tumor stages when compared to *CIC-ERF* replete tumors (*Figure 1D*). In order to understand the association between *CIC* and/or *ERF* alterations in specific PCa cohorts, we stratified published datasets to identify patients that represent primary PCa (*Fraser et al., 2017*; *Hieronymus et al., 2014*; ' *Cancer Genome Atlas Research Network, 2015*) (PNAS 2014, n=272; Cell 2015, n=333; Nature 2017, n=477 primary PCas) and metastatic castrate resistant prostate cancer (mCRPC) (*Abida et al., 2019*; *Grasso et al., 2012*; *Robinson et al., 2015*) (Nature 2012, n=50; Cell 2015, n=150; PNAS 2019, n=429 mCRPCs). This analysis revealed enrichment of *CIC* and *ERF* alterations including the *CIC-ERF* co-deletion in mCRPC samples (*Figure 1E*, *Figure 1—source data 1*, *Figure 1—figure supplement 1*). Importantly, *CIC-ERF* co-deleted tumors clustered as a subgroup when compared to the more well-characterized molecular subsets including *ERG*, *ETV1*, *ETV4*, *SPOP*, and *FOXA1* altered PCas, suggesting a distinct molecular subtype of PCa (*Figure 1F*). In order to explore clinical outcomes of patients harboring *CIC-ERF* co-deleted tumors, we performed a survival analysis using the aforementioned PCa datasets and observed significantly worse outcomes in patients who harbored the *CIC-ERF* co-deletion (p=0.001, disease-free survival [DFS] [*ERF-CIC* co-deletion 25 events/90 total; no *ERF-CIC* co-deletion 153 events/910 total] and p=0.01, progression-free

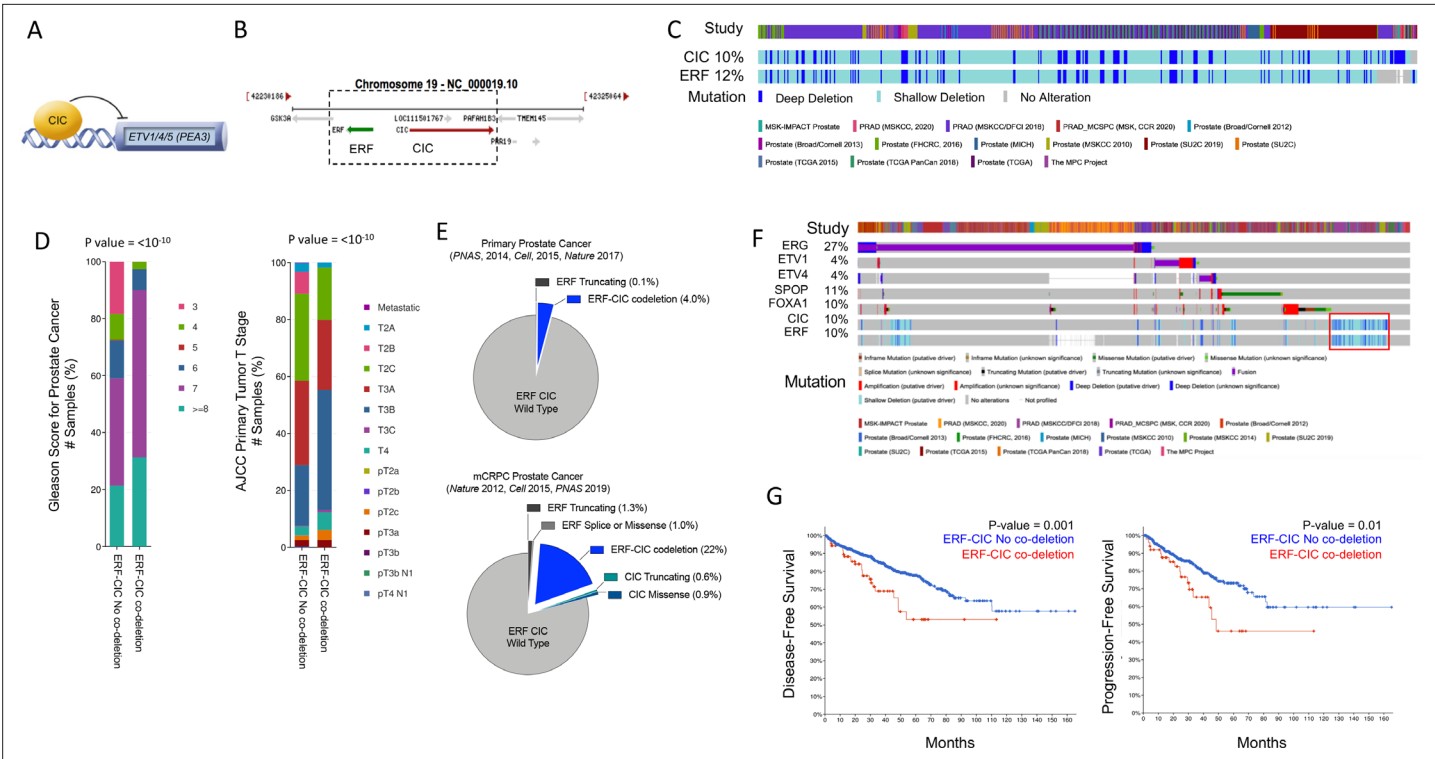

**Figure 1.** Capicua (*CIC*) and ETS2 repressor factor (*ERF*) are co-deleted in aggressive prostate cancer (PCa) and associate with worse clinical outcomes. (**A**) CIC transcriptionally represses ETV1/4/5. (**B**) The 19q13.2 genomic locus demonstrating the physical location of *ERF* and *CIC*. (**C**) 15 PCa studies (cBioPortal) demonstrating the co-occurrence of *ERF* and *CIC* homozygous and heterozygous deletions. The co-occurrence of *ERF* and *CIC* alterations is highly significant (p<0.001 co-occurrence, Fisher exact test). (**D**) *ERF-CIC* co-deleted PCa stratified by Gleason score and tumor stage. (**E**) Frequency of *ERF* and *CIC* alterations in primary PCa (top) and metastatic castrate resistant prostate cancer (mCRPC; bottom), demonstrating enrichment in mCRPC. (**F**) Onco-print of known genetic drivers (ERG, ETV1, ETV4, SPOP, and FOXA1) of PCa aligned with *CIC* and *ERF* (cBioPortal). *CIC-ERF* co-deleted prostate tumors (red box) do not frequently co-occur with other known oncogenic events. (**G**) Survival analysis performed using 15 PCa datasets from cBioPortal. Disease-free survival (DFS) and progression-free survival (PFS) in patients harboring the *ERF-CIC* co-deletion (red) vs. no *ERF-CIC* co-deletion (blue). p=value, log-rank.

The online version of this article includes the following source data and figure supplement(s) for figure 1:

**Source data 1.** Prostate cancer studies identified in cBioPortal demonstrating the total number of patients, number of patients with shallow or deep deletions in Capicua (CIC)-ETS2 repressor factor (ERF), and the frequency of *CIC-ERF* alterations in each cohort.

**Figure supplement 1.** ETS2 repressor factor (*ERF*)-Capicua (*CIC*) co-deletion frequency across 15 prostate cancer studies.

survival [PFS] [*ERF-CIC* co-deletion 16 events/52 total; no ERF-CIC co-deletion 77 events/442 total]; *Figure 1G*). One limitation of this analysis is that individual studies included overlapping tumors from the same patient (i.e. TCGA PanCancer and Firehose Legacy cohorts). Despite this overlap, we utilized this insight to formulate a hypothesis that *CIC* and *ERF* are co-deleted with increasing frequency in mCRPC and that the *CIC-ERF* co-deletion is associated with worse clinical outcomes in PCa patients.

Our initial findings provided rationale to explore the genetic and functional relationship between CIC and ERF. Independently, CIC and ERF have previously been reported to promote malignant phenotypes, including tumor growth and metastasis in multiple human cancer subsets (*Bettegowda et al., 2011*; *Bose et al., 2017*; *Choi et al., 2015*; *Huang et al., 2017*; *Kawamura-Saito et al., 2006*; *Kim et al., 2018*; *Bunda et al., 2019*; *Okimoto et al., 2019*; *Okimoto et al., 2017*; *Simón-Carrasco et al., 2017*; *Wong et al., 2019*; *Yang et al., 2017*). Since our clinical data indicated that combinatorial loss of *CIC* and *ERF* was associated with worse patient outcomes, we hypothesized that *CIC* and *ERF* loss may cooperate to enhance PCa progression. To investigate this, we engineered *ERF*, *CIC*, or both *CIC* and *ERF*-deficient immortalized prostate epithelial cells (PNT2) and performed a series of *in vitro* and *in vivo* experiments to test the combinatorial effect of the *CIC-ERF* co-deletion. Compared to single gene loss of *CIC* or *ERF*, genetic inactivation of both *CIC* and *ERF* increased colony formation (*Figure 2A–B*, *Figure 2—figure supplement 1A*) and spheroid formation (*Figure 2C–D*) in PNT2

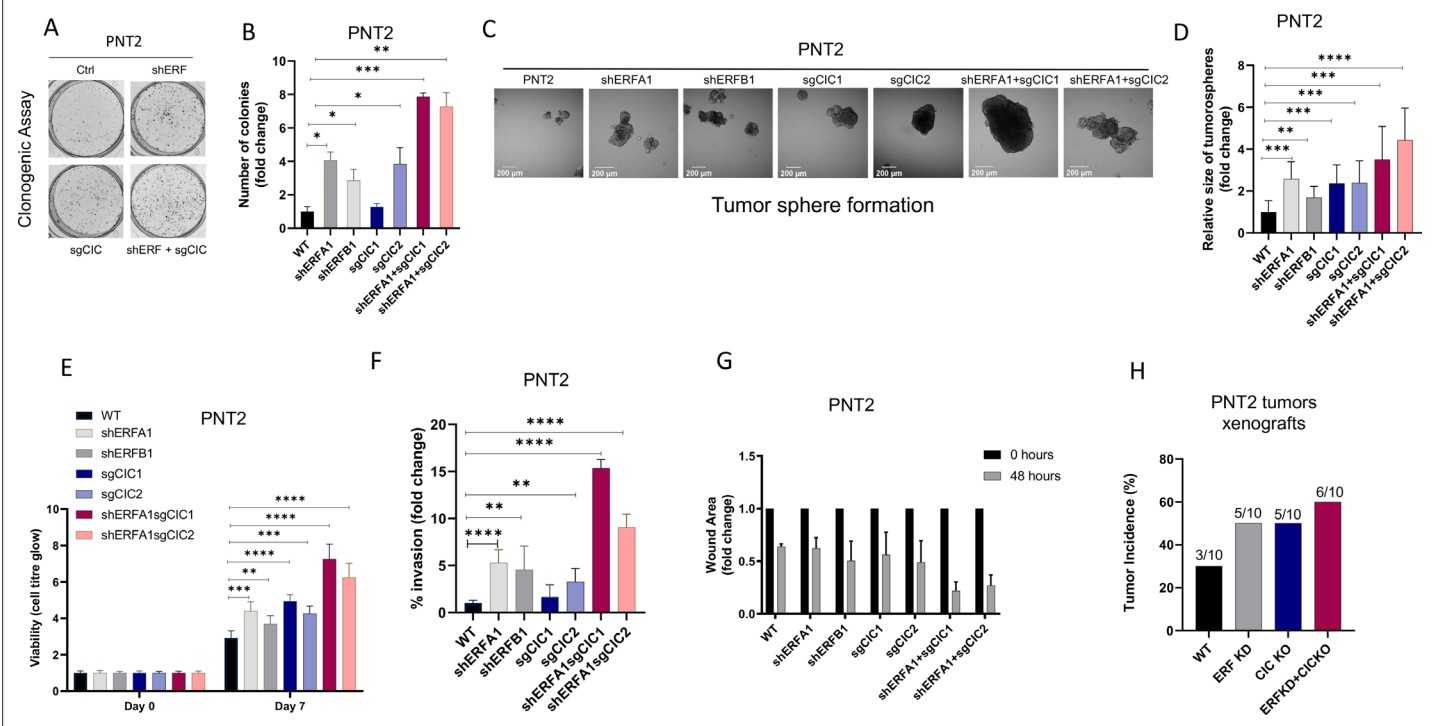

**Figure 2.** Capicua (*CIC*) and ETS2 repressor factor (*ERF*) loss promote tumor formation and control malignant potential in prostate epithelial cells. (**A**) Clonogenic assay comparing prostate epithelial cells (PNT2) with *ERF* KD, *CIC* KO, or *ERF* KD+*CIC* KO compared to control. (**B**) Number of colonies for each condition in (**A**) (n=3). (**C**) Spheroid growth assay using PNT2 cells expressing *ERF* KD, *CIC* KO, *ERF* KD+*CIC* KO vs. control. (**D**) Size of the sphere for each condition in (**C**) (n=6). Error bars represent SD. p Values were calculated using Student's t test. *p<0.05, **p<0.01, ***p<0.001, and ****p<0.0001. (**E**) Cell-titer glo viability assay (n=6), (**F**) transwell assay (n=3), and (**G**) wound healing assay comparing PNT2 *ERF* KD, *CIC* KO, and *ERF* KD+*CIC* KO to control (n=4). Error bars represent SD. p Values were calculated using Student's t test. (**H**) Bar graph comparing the incidence of PNT2 parental (N=3/10), PNT2 *ERF* KD (5/10), PNT2 *CIC* KO (N=5/10), or PNT2 *ERF* KD+CIC KO (N=6/10) tumor formation in immunodeficient mice. **p<0.01, ***p<0.001, and ****p<0.0001.

The online version of this article includes the following source data and figure supplement(s) for figure 2:

**Figure supplement 1.** Capicua (CIC) and ETS2 repressor factor (ERF) loss enhances tumor formation in prostate epithelial cells (PNT2).

**Figure supplement 1—source data 1.** Full length western blot images of Capicua (CIC), ETS2 repressor factor (ERF) and HSP90 in prostate epithelial cells (PNT2) and its variants with associated raw images.

cells. *CIC-ERF* co-deletion also enhanced malignant phenotypes including cellular viability, invasiveness, and migratory capacity in PNT2 (*Figure 2E–G*). Additionally, genetic silencing of both *CIC* and *ERF* increased the frequency of subcutaneous tumor xenograft formation in severe-combined immunodeficient (SCID) mice compared to control (*Figure 2H*, *Figure 2—figure supplement 1B*). Thus, our findings demonstrate that the combination of *CIC* and *ERF* loss augments the transformation of PNT2 and promotes malignant phenotypes.

To assess the functional role of CIC and ERF in the context of human PCa progression, we leveraged two genetically annotated, androgen-insensitive PCa cell lines, DU-145 (moderate metastatic potential) and PC-3 (high-metastatic potential). DU-145 cells harbor a loss-of-function ERF mutation (*ERF^{A132S}*) (*Huang et al., 2017*) and express functional WT *CIC*, (*Figure 3—figure supplement 1A*). By comparison, PC-3 cells are deficient in *CIC* (homozygous deletion) and retain functional ERF (*Figure 3—figure supplement 1B*; *Cerami et al., 2012*; *Gao et al., 2013*; *Seim et al., 2017*). Thus, these cell line models provided isogenic systems to functionally interrogate the role of CIC and ERF in human PCa. Specifically, genetic reconstitution of *ERF* into *ERF*-deficient DU-145 cells decreased colony formation in both *CIC* proficient (parental cells) and *CIC* knockout (KO) conditions (*Figure 3A–B*, *Figure 3—figure supplement 1C, D*). While *CIC* loss did not enhance colony formation in *ERF*-deficient DU-145 cells, it significantly increased tumor cell viability, invasion, and migratory capacity compared to control (*Figure 3C–D*, *Figure 3—figure supplement 1E*). Importantly, we observed that ERF expression in *CIC* KO DU-145 cells rescued the CIC-mediated effects on viability

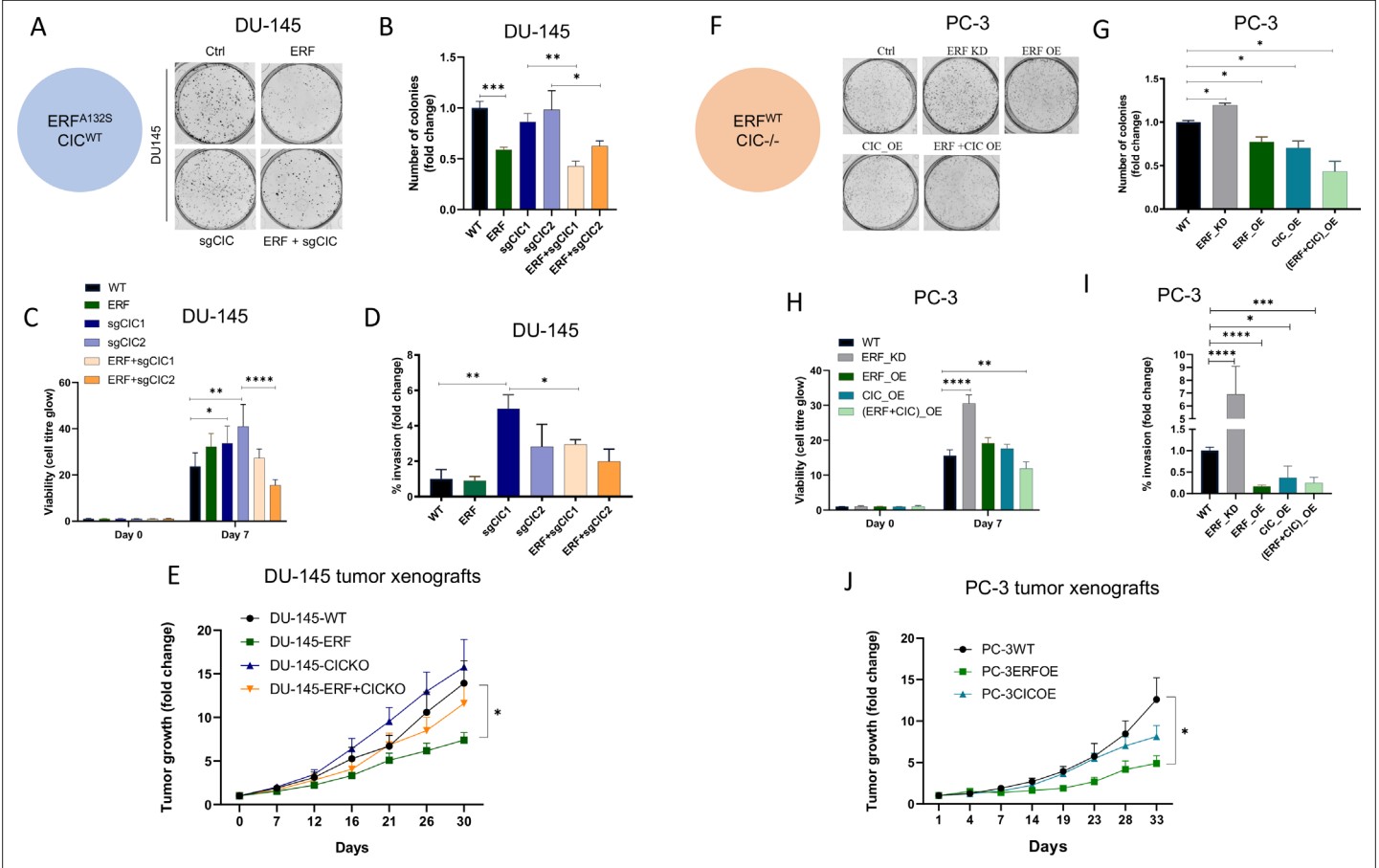

**Figure 3.** Capicua (CIC) and ETS2 repressor factor (ERF) mutually suppress malignant phenotypes in human prostate cancer (PCa). (**A**) Clonogenic assay of DU-145 cells with *ERF* rescue, *CIC* knockout (KO), or *ERF* rescue +*CIC* KO compared to parental control. (**B**) Number of colonies for each condition in (**A**) (n=3). (**C**) Cell-titer glo viability assay (n=6) and (**D**) transwell assay comparing DU-145 parental cells to DU-145 with *ERF* rescue, *CIC* KO, or *ERF* rescue +CIC KO (n=3). p Values were calculated using Student's t test. *p<0.05, **p<0.01, ***p<0.001, and ****p<0.0001. Error bars represent SD. (**E**) Relative tumor volume in mice bearing DU-145 parental, DU-145 *ERF*, DU-145 with *CIC* KO, or DU-145 *ERF* +*CIC* KO xenografts (N=10). p Values were calculated using Student's t test. *p<0.05. Error bars represent SEM. (**F**) Clonogenic assay in PC-3 cells expressing *ERF* knockdown (KD), *ERF* overexpression (OE), *CIC* OE, or *ERF* +*CIC* OE compared to control. (**G**) Number of colonies for each condition in (**F**) (n=3). (**H**) Cell-titer glo viability assay (n=6) and (**I**) transwell assay comparing different groups in PC-3 cells (WT, *ERF* KD, *ERF* OE, *CIC* OE, or *ERF* +*CIC* OE) (n=3). p Values were calculated using Student's t test. *p<0.05, **p<0.01, ***p<0.001, and ****p<0.0001. Error bars represent SD. (**J**) Relative tumor volume in mice bearing PC-3 parental cells, PC-3 *ERF* OE, or PC-3 *CIC* OE (N=10) over 33 days. p Values were calculated using Student's t test. *p<0.05. Error bars indicate SEM.

The online version of this article includes the following source data and figure supplement(s) for figure 3:

**Figure supplement 1.** DU-145 and PC-3 prostate cancer cells are well defined model systems to study Capicua (CIC) and ETS2 repressor factor (ERF) function.

**Figure supplement 1—source data 1.** Full-length western blot images of basal levels of Capicua (CIC), ETS2 repressor factor (ERF), and β-actin in prostate epithelial cells (PNT2), DU145, and PC3 cells and associated raw images.

**Figure supplement 1—source data 2.** Full-length western blot images of CIC, ETS2 repressor factor (ERF), and HSP90 in DU145 cells with its variants and associated raw images.

**Figure supplement 1—source data 3.** Full-length western blot images of ETS2 repressor factor (ERF) and HSP90 in PC3 cells with its variants and associate raw images.

and migration/invasion (*Figure 3C–D*, *Figure 3—figure supplement 1E*). These findings indicate that rescuing ERF can partially restore the functional effects of CIC loss in DU-145 cells. To further understand if ERF could suppress tumor growth *in vivo*, we reconstituted WT *ERF* into *CIC* proficient and deficient (*CIC* KO) DU-145 cells and generated subcutaneous xenografts in immunodeficient mice (NU/J). Consistent with our *in vitro* data, *CIC* KO increased the tumor growth rate *in vivo* and genetic reconstitution of WT *ERF* partially suppressed tumor growth compared to DU-145 *CIC* KO cells

(*Figure 3E*). Moreover, genetic reconstitution of *ERF* into *CIC* proficient DU-145 cells suppressed the tumor growth rate *in vivo* (*Figure 3E*). We next used PC-3 cells, to further test how ERF and CIC functionally interact in the context of human PCa. We first noted that *ERF* overexpression (OE) or reconstitution of CIC alone decreased PC-3 colony formation, with combinatorial *ERF* OE and *CIC* rescue having the most significant reduction compared to parental PC-3 cells (*Figure 3F–G*, *Figure 3—figure supplement 1F, G*). Moreover, *ERF* and *CIC* expression had a similar impact on decreasing PC-3 viability, invasion, and migratory capacity (*Figure 3H–I*, *Figure 3—figure supplement 1H*). Similarly, genetic suppression of *ERF* resulted in a significant increase of colony formation, viability, and invasion compared to parental PC3 cells (*Figure 3F–I*, *Figure 3—figure supplement 1I*). Interestingly, OE of ERF in mice bearing *CIC* deficient PC-3 tumor xenografts decreased tumor growth compared to PC-3 parental and PC-3 cells expressing WT *CIC* (genetic rescue of CIC; *Figure 3J*). Since we consistently observed that *ERF* expression could partially rescue the effects of *CIC* loss in PCa, we hypothesized that WT CIC and ERF potentially cooperate to limit PCa progression.

In order to mechanistically define how CIC and ERF (two TFs with known repressor function) were interacting to functionally regulate PCa, we performed chromatin immunoprecipitation followed by sequencing (ChIP-Seq) using a validated CIC antibody (*Lin et al., 2020*; *Okimoto et al., 2019*; *Okimoto et al., 2017*) in PNT2, and compared this to a publicly available ERF ChIP-Seq dataset in VCaP PCa cells (*Bose et al., 2017*), we were unsuccessful at pulling down ERF in PNT2 cells. This analysis identified 178 high-confidence (False-discovery rate (FDR) ≤0.05) CIC peaks that mapped to 130 annotated genes including known targets, *ETV1*, *ETV4*, and *ETV5*. Globally, CIC peaks were localized to distinct genomic regions including promoters (38.6%), untranslated regions (UTRs) (1.14%), introns (19.9%), and distal intergenic regions (38.64%). Interestingly, the distribution of ERF peaks was similar to CIC, with 32.8% in promoters, 1.2% in UTRs, 29.2% intronic, and 33.8% in distal intergenic regions (*Figure 4A*). Next, through a comparative ChIP-Seq analysis (*Figure 4B*), we identified 91 shared CIC and ERF target genes. Importantly, we focused on genes with shared CIC- and ERF-binding sites to potentially explain the functional cooperativity that we observed in our prior studies. In order to narrow down potential candidates, we performed Functional Clustering Analyses through the DAVID bioinformatics database (https://david.ncifcrf.gov) (*Huang et al., 2009*; *Sherman et al., 2022*) using the 91 shared CIC and ERF target genes (*Supplementary file 1*). Among these putative CIC and ERF targets, the *PEA3* (*ETV1*, *ETV4*, and *ETV5*) TFs (known oncogenic drivers in PCa; *Baena et al., 2013*; *Kim et al., 2016*; *Oh et al., 2012*) were found to be the most highly enriched family. These findings suggested that CIC and ERF may co-regulate PEA3 family members through direct transcriptional control.

In order to further identify how CIC and/or ERF impact *PEA3* TF expression, we performed quantitative real time polymerase chain reaction (qRT-PCR) in PNT2 cells, assessing *ETV1*, *ETV4*, or *ETV5* mRNA levels in response to genetic silencing of *CIC* and/or *ERF*. As expected, *CIC* KO and/or combinatorial *CIC* and *ERF* loss in PNT2 cells (*ERF* and *CIC* WT) consistently increased *ETV1*, *ETV4*, and *ETV5* levels compared to control (*Figure 4—figure supplement 1A-F*). In contrast, genetic silencing of *ERF* consistently increased *ETV1* mRNA expression, but not *ETV4* or *ETV5* (*Figure 4C*, *Figure 4—figure supplement 1G-J*). These findings indicated that *ETV1* may be a shared transcriptional target of CIC and ERF in prostate cells. In order to test a potential proteomic interaction between CIC and ERF, we performed co-immunoprecipitation (Co-IP) experiments but did not observe binding when either ERF or CIC was pulled down in HEK293T cells (*Figure 4—figure supplement 1K-L*). This led us to hypothesize that CIC and ERF may potentially bind (but not interact) to distinct regions along the *ETV1* regulatory element. To explore this, we first localized CIC (TGAATGGA) and ERF (GGAA) DNA binding motifs within the proximal upstream regulatory element of *ETV1* and independently confirmed CIC and ERF occupancy of the *ETV1* promoter through ChIP-PCR (*Figure 4D–G*, *Figure 4—figure supplement 1M, P*). To extend these findings into the context of PCa, we reconstituted *ERF* in *ERF* deficient DU-145 cells and this consistently decreased *ETV1* expression (not *ETV4* or *ETV5*) in both CIC proficient and CIC deficient settings (*Figure 4H*, *Figure 4—figure supplement 1Q-T*). Moreover, *ERF* Knockdown (KD) or *ERF* OE in *CIC* deficient PC-3 cells increased and decreased *ETV1* mRNA expression, respectively (*Figure 4I–J*). Since *ETV1* is a known target of CIC (*Dissanayake et al., 2011*; *Jiménez et al., 2012*), we focused on further validating *ETV1* as a molecular target of ERF. To this end, we engineered a *ETV1* luciferase based promoter assays and observed a decrease in luciferase activity following ERF expression in 293T cells (*Figure 4K*). These genetic tools further validate that

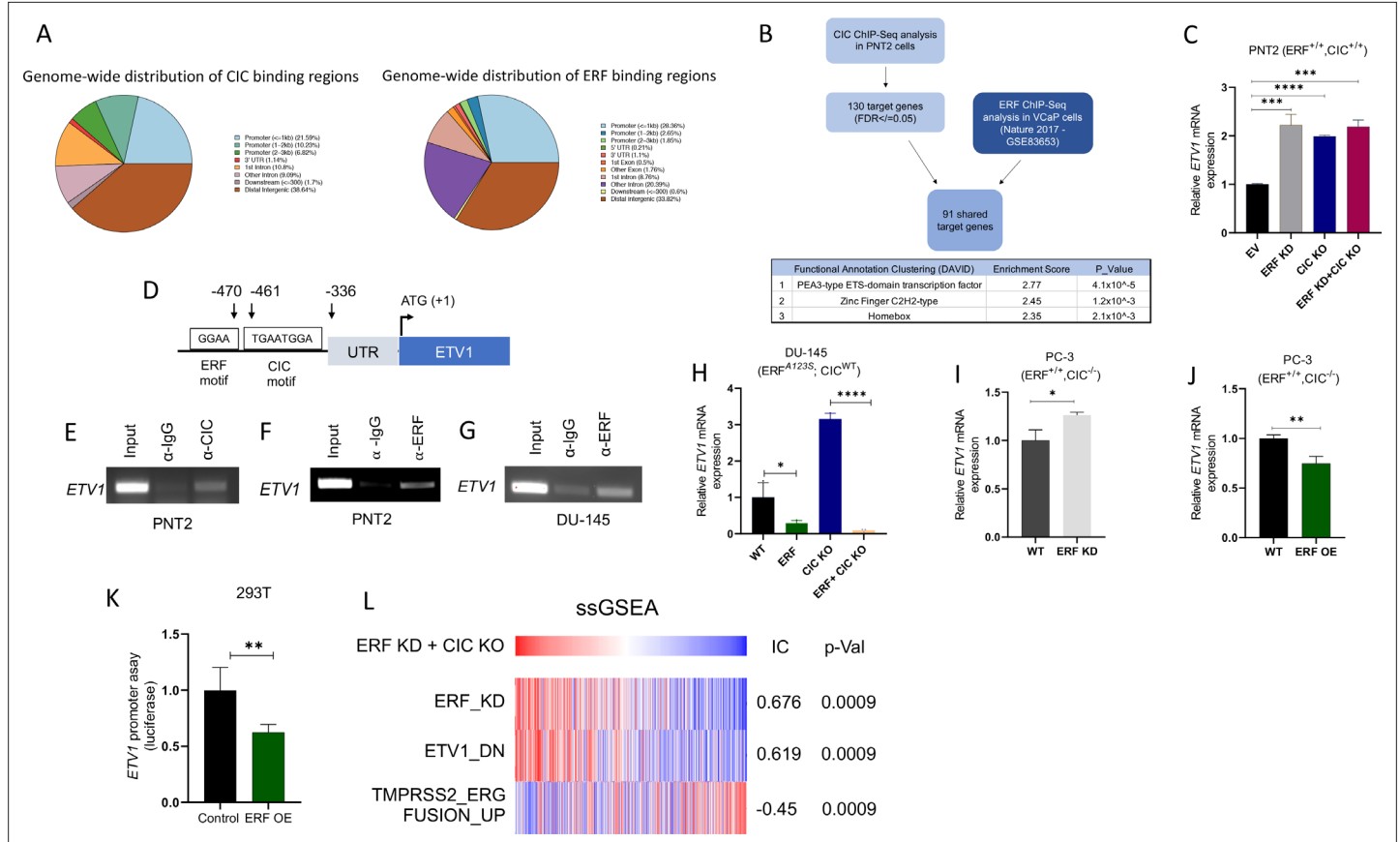

**Figure 4.** Capicua (CIC) and ETS2 repressor factor (ERF) cooperatively bind an *ETV1* regulatory element to suppress *ETV1* expression and transcriptional activity. (**A**) Percentage of CIC and ERF peaks located in defined genomic regions. (**B**) Schematic algorithm to identify shared CIC and ERF target genes in prostate cells (top). Functional Clustering Analysis of the 91 shared CIC and ERF target genes using DAVID (bottom table). (**C**) *ETV1* mRNA expression in prostate epithelial cells (PNT2) (*CIC-ERF-replete*) cells with *ERF* knockdown (KD), *CIC* knockout (KO), or *ERF* KD +*CIC* KO (n=3). (**D**) Schematic of CIC and ERF DNA-binding motifs in the *ETV1* promoter. (**E**) Chromatin immunoprecipitation (ChIP)-PCR from PNT2 cells showing CIC occupancy on the *ETV1* promoter. (**F–G**) ChIP-PCR with ERF occupancy on the *ETV1* promoter. (**H**) *ETV1* mRNA expression in DU-145 (ERF-deficient) cells with *ERF* rescue, *CIC* KO, or *ERF* rescue +*CIC* KO (n=3). *ETV1* mRNA expression in PC-3 cells with (**I**) *ERF* KD (n=3) and (**J**) *ERF* overexpression (OE) (n=3). p Values were calculated using Student's t test. *p<0.05, **p<0.01, and ****p<0.0001. Error bars represent SD. Performed in triplicate. (**K**) *ETV1* luciferase promoter assay in 293T cells comparing Empty vector (EV) with *ERF* OE (n=6). Student's t test, *p<0.05. Error bars represent SD. (**L**) Single sample gene set enrichment analysis (ssGSEA) alignments comparing gene expression patterns in PNT2 cells with *ERF* KD and *CIC* KO. IC = information coefficient.

The online version of this article includes the following source data and figure supplement(s) for figure 4:

**Source data 1.** Full-length PCR gel images of *ETV1* after Capicua (CIC) pull down in prostate epithelial cells (PNT2).

**Source data 2.** Full-length PCR gel images of *ETV1* after ETS2 repressor factor (ERF) pull down in prostate epithelial cells (PNT2).

**Source data 3.** Full-length PCR gel images of *ETV1* after ETS2 repressor factor (ERF) pull down in DU-145 cells.

**Figure supplement 1.** ETV1, but not ETV4 or ETV5, is a transcriptional target of both ETS2 repressor factor (ERF) and Capicua (CIC).

**Figure supplement 1—source data 1.** Co-immunoprecipitation using GFP-tagged ETS2 repressor factor (ERF) and immunoblotting for Capicua (CIC; bottom panel) and ETS2 repressor factor (ERF; top panel) with associated raw images.

**Figure supplement 1—source data 2.** Co-immunoprecipitation using myc-tagged Capicua (CIC) and immunoblotting for ETS2 repressor factor (ERF; top panel) and CIC (bottom panel) with associated raw images.

ERF can suppress *ETV1* expression through direct transcriptional silencing of the *ETV1* promoter and identifies *ETV1* as an ERF target.

Consistent with a repressor function, ERF loss was previously shown to transcriptionally associate with ETV1-regulated gene set signatures (***Huang et al., 2017***). Yet it remains unclear if ERF can directly regulate *ETV1* and how combinatorial loss of CIC and ERF controls ETV1-mediated (or other ETS family members) transcriptional programs. To explore this, we generated a signature gene set of upregulated genes from our dual *CIC* and *ERF* deficient PNT2 cells (*ERF* KD +*CIC* KO) and projected

the Cancer Genome Atlas PCa (TCGA-PRAD) dataset onto the transcriptional space of these signature gene sets using the ssGSEA module (Version 10.0.9) on GenePattern (*Reich et al., 2006*). We found that *CIC* and *ERF* loss were significantly associated with the *ETV1*-regulated gene set (Information coefficient (IC)=0.619, p=0.0009), but was anti-correlated with the *TMPRSS2-ERG* fusion signature gene set (*Setlur et al., 2008*; *Figure 4L*, IC = –0.45, p=0.0009). Thus, the enrichment of the *ETV1*-regulated gene set signature was shared between *ERF* loss alone (*Huang et al., 2017*) and *CIC-ERF* dual suppression (*Figure 4L*). In contrast, combinatorial *CIC* and *ERF* loss negatively correlated with the *TMPRSS2-ERG* fusion signature gene set, which was not consistent with our prior studies using *ERF* KD alone (*Huang et al., 2017*). These findings led us to hypothesize that the dual suppression of *CIC* and *ERF* may increase ETV1-mediated transcriptional programs in PCa. This was supported by two major lines of experimental and conceptual evidence including: (1) dual suppression of *ETV1* expression and ETV1 mediated transcriptional output by CIC and ERF; and (2) the majority of tumors derived from PCa patients that harbor *ERF* deletions also contain deletions in *CIC*.

In order to demonstrate enhanced survival dependence on ETV1 in PCa cells, we genetically silenced *CIC* in ERF-deficient DU-145 cells and assessed drug sensitivity to an ETV1 inhibitor (BRD32048) (*Pop et al., 2014*). We validated the pharmacologic effect of ETV1 inhibition (BRD32048) through suppression of known downstream target genes (*ATAD2* and *ID2)* using qPCR in LNCaP cells (*Figure 5—figure supplement 1A-C*). Since BRD32048 was previously shown to decrease invasiveness in ETV1-fusion positive PCa cells (LNCaP), but not significantly impact tumor cell viability, we unexpectedly observed that silencing *CIC* in DU-145 cells mildly enhanced sensitivity to BRD32048 (*Figure 5A*). To further confirm these findings and to mitigate potential off-target effects, we silenced *ETV1* using siRNA in PC3 and DU-145 cells expressing Crispr-based sgRNA targeting *CIC* (*Figure 5—figure supplement 1D-F*). Consistent with our pharmacologic studies, the viability of DU-145 *CIC* KO cells was decreased upon genetic *ETV1* inhibition (*Figure 5B*). Similarly, pharmacologic and genetic ETV1 inhibition decreased invasiveness of CIC deficient DU-145 cells (*Figure 5C–D*). These findings indicate that loss of CIC in ERF deficient PCa cells can potentially modulate the sensitivity to ETV1-directed therapies. To expand these findings, we also consistently observed a decrease in viability and invasiveness upon ETV1 inhibition (BRD32048) in PNT2 cells with dual *ERF* and *CIC* suppression and in PC-3 cells with *ERF* KD (*Figure 5—figure supplement 2A-J*). These *in vitro* findings indicate that ETV1 inhibition in *CIC* and *ERF* deficient prostate cells can suppress invasion and potentially limit viability. Further studies targeting ETV1 in patient derived specimens that harbor endogenous *CIC-ERF* co-deletions is warranted. Collectively, our data indicate that CIC and ERF may cooperate to silence ETV1 transcriptional programs, limiting ETV1-mediated PCa progression.

## Discussion

Molecular and functional subclassification of human PCa has revealed a dependence on ETS family TFs including ERG, ETV1, ETV4, and ETV5 (*Feng et al., 2014*; *Oh et al., 2012*). The predominant mode of ETS activation in PCa is through chromosomal rearrangements that fuse *ERG, ETV1, ETV4, and ETV5* to the androgen-regulated *TMPRSS2* gene, leading to fusion oncoproteins that drive oncogenesis (*Clark and Cooper, 2009*; *Feng et al., 2014*; *Helgeson et al., 2008*; *Tomlins et al., 2006*; *Tomlins et al., 2005*). Interestingly, recent data indicate that *ETV1, ETV4,* and *ETV5* are upregulated in a fusion-independent manner and are associated with poor clinical outcomes in PCa patients (*Baena et al., 2013*; *Hermans et al., 2008*). Our study focused on understanding the molecular mechanisms that drive fusion-independent upregulation of ETS family members and we reveal a new molecular subclass of PCa defined by a co-deletion of two TFs, *CIC* and *ERF*.

CIC is a TF that directly silences *ETV1, ETV4, and ETV5* transcription through direct repression at proximal regulatory sites (*Jiménez et al., 2012*). We observed that in ~10–12% of human PCa, *CIC* and *ERF* are co-deleted through focal homozygous or heterozygous deletions. It has been recently shown that ERF competes for ETS DNA binding motifs and our studies identify cooperative regulation of key target genes between CIC and ERF. Specifically, through ChIP-Seq analysis coupled with a series of *in vitro* and *in vivo* studies, we identify a coordinated binding of CIC and ERF to the proximal *ETV1* regulatory element that physically and functionally regulates ETV1 expression. Therefore, we reveal *ETV1* as a novel ERF target gene. Rescuing *ERF* in *CIC* deficient PCa cells decreases *ETV1* expression and limits malignant phenotypes including viability, migration and invasion.

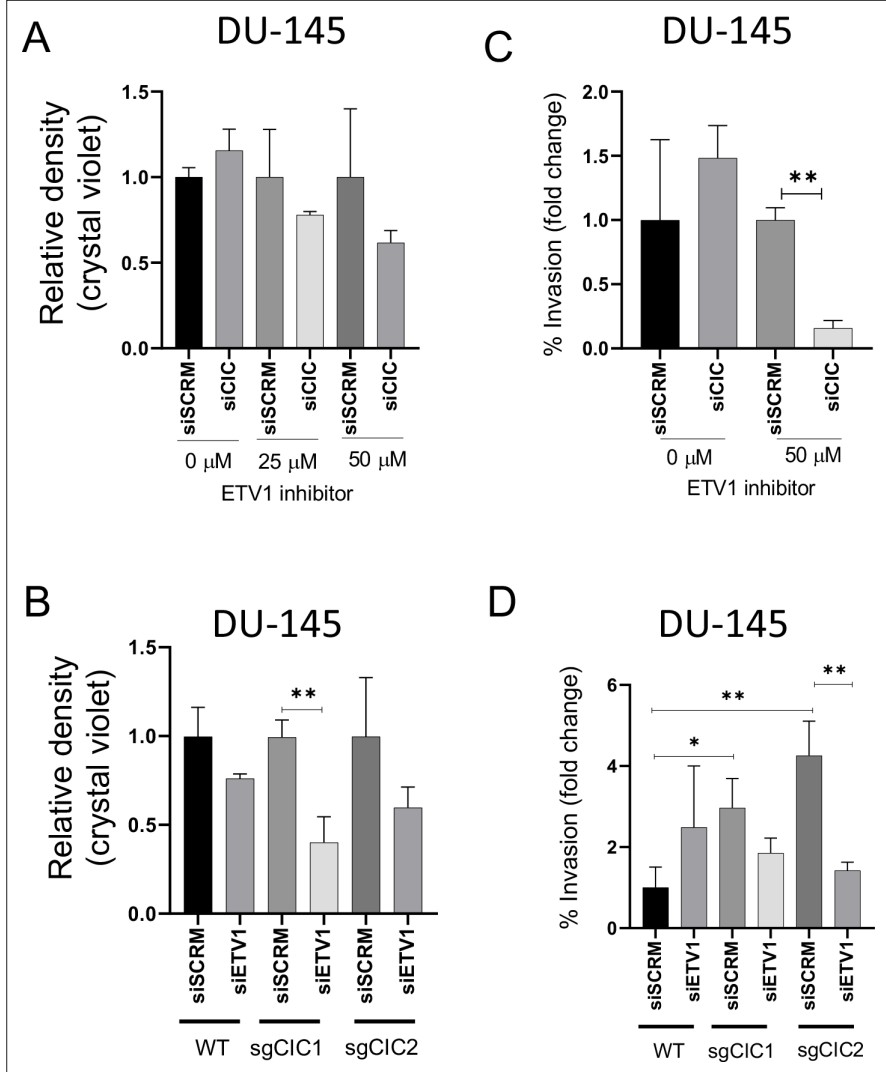

**Figure 5.** Combinatorial CIC and ETS2 repressor factor (ERF) loss can modulate ETV1 inhibitor sensitivity in prostate cells. (**A**) DU-145 cells were transfected with either siScramble (siSCM) or si*CIC*. After 48 hr, BRD32048 (ETV1 inhibitor) was added to both the transfected groups at the defined concentrations (0 μM, 25 μM, 50 μM). After 24 hr of BRD32048 treatment, cells were replated for crystal violet assay (0.4%) and images were taken and analyzed after 5 days (n=3). siCIC was compared to siSCRM conditions in each respective drug concentration. (**B**) Crystal violet viability assay (n=3). si*ETV1* groups were compared to siSCRM control groups +/- *CIC* expression (sgCtrl, sgCIC1, or sgCIC2). (**C**) DU-145 cells were transfected with either siScramble (siSCM) or si*CIC*. After 48 hr, BRD32048 was added to the transfected groups at defined concentrations (0 μM or 50 μM). Transwell invasion assays (n=3) were performed 24 hr after the addition of BRD32048. siCIC was compared to siSCRM in the 0 μM and 50 μM concentration groups. (**D**) Transwell invasion assays (n=3) comparing si*ETV1* to siSCRM control +/-CIC expression (sgCtrl, sgCIC1, or sgCIC2). p value = *p<0.05, **p<0.01 for all figures. Error bars represent SD.

The online version of this article includes the following figure supplement(s) for figure 5:

**Figure supplement 1.** Validation of ETV1 chemical (BMS32048) and genetic inhibition in prostate cancer cells.

**Figure supplement 2.** CIC and ETS2 repressor factor (ERF) expression modulate sensitivity to ETV1 inhibitor, BRD32048.

The 19q13.2 locus contains *CIC* and *ERF*, which are physically adjacent and oriented in opposing directions. The long and short isoforms of *CIC* are separated from *ERF* by ~15 and 30 kb, respectively. Thus, future studies directed at defining the genome topology and epigenetic states, both within and around this highly conserved region are warranted. In particular, studies aimed at mapping topology associated domains and key histone marks can potentially reveal shared upstream regulatory elements

including enhancers or superenhancers that may co-regulate *CIC* and *ERF* in concert. These findings could reveal non-genetic mechanisms to functionally regulate *CIC* and *ERF* expression in a coordinated fashion.

Beyond PCa, we and others have observed that *CIC* and *ERF* are co-deleted in a subset of stomach adenocarcinoma (*LeBlanc et al., 2017*; *Okimoto et al., 2017*). Thus, combined *CIC* and *ERF* loss are not entirely specific for PCa. We speculate that since prostate (*Baena et al., 2013*) and potentially stomach adenocarcinoma (*Keld et al., 2011*) are highly dependent on PEA3 transcriptional dysregulation to enhance tumor progression, the dual loss of *CIC* and *ERF* may, in part, represent an alternative mode to de-repress ETS mediated transcriptional programs in these cancer subsets.

The use of ETV1 inhibitors has been limited to preclinical studies (*Pop et al., 2014*). These studies have largely focused on direct targeting of ETV1 fusion oncoproteins in PCa (*Pop et al., 2014*). Our findings indicate that patients with *CIC-ERF* deficient PCa may have a survival dependence on ETV1 and that these patients may potentially benefit from ETV1 directed therapies. A limitation of this targeted approach in PCa patients is that we did not find a statistically significant difference in *ETV1* transcript levels in *CIC-ERF* co-deleted tumors compared to *CIC-ERF* replete tumors in the TCGA-PRAD (n=455) dataset. One potential explanation is that *ETV1* is commonly upregulated in human PCas through mechanisms beyond *CIC-ERF* loss. Thus, while our data support the upregulation and potential induced dependence on ETV1 in our *CIC-ERF* deficient systems, it remains unclear if this will translate beyond our cell-line based models into PCa patients that harbor *CIC-ERF* co-deletions. Thus, larger studies that aim to evaluate: (1) *ETV1* mRNA and protein expression levels; (2) the biological significance of ETV1 function; and (3) the clinical application of ETV1 inhibitors in patients with endogenous *CIC-ERF* co-deleted tumors (compared to *CIC-ERF* WT tumors) is warranted in this subset of PCa. Collectively, we have uncovered a molecular subset of PCa defined by a co-deletion of *CIC* and *ERF* and further demonstrate a mechanism-based strategy to potentially limit tumor progression through ETV1 inhibition in this subset of human PCa.

# Materials and methods

## Key resources table

| Reagent type (species) or resource | Designation | Source or reference | Identifiers | Additional information |
|---|---|---|---|---|
| Gene CIC (*Homo sapiens*) | *CIC* | Pubmed gene database | Gene ID: 23152 | |
| Gene ERF (*Homo sapiens*) | *ERF* | Pubmed gene database | Gene ID: 2077 | |
| Gene ETV1 (*Homo sapiens*) | *ETV1* | Pubmed gene database | Gene ID: 2115 | |
| Cell line (*Homo-sapiens*) | HEK293T | ATCC | | Cell line maintained in DMEM with 10% FBS and 1% PSN |
| Cell line (*Homo-sapiens*) | PNT2 | Sigma | | Cell line maintained in DMEM with 10% FBS and 1% PSN |
| Cell line (*Homo-sapiens*) | DU-145 | ATCC | | Cell line maintained in RPMI with 10% FBS and 1% PSN |
| Cell line (*Homo-sapiens*) | PC3 | ATCC | | Cell line maintained in RPMI with 10% FBS and 1% PSN |
| Transfected construct (human) | ERF shRNA #1 | Sigma-Aldrich | CAT# TRCN000001391 TRCN0000013912 | Lentiviral construct to transfect and express ERF shRNA. |
| Transfected construct (human) | sgRNAs | Addgene | CAT#74959 and #74953 | |
| Transfected construct (human) | Lentiviral GFP-tagged ERF | GeneCopoeia | CAT# EX-S0501-Lv122 | |
| Transfected construct (human) | CIC-Myc-tag plasmid | Origene | CAT#: RC215209 | |

*Continued on next page*

*Continued*

| Reagent type (species) or resource | Designation | Source or reference | Identifiers | Additional information |
|---|---|---|---|---|
| Transfected construct (human) | siRNA to ETV1 SMARTpool | Dharmacon | CAT# L-003801-00-0005 | |
| Transfected construct (human) | siRNA to CIC SMARTpool | Dharmacon | CAT# L-015185-01-0005 | |
| Antibody | CIC (Rabbit polyclonal) | Thermo Fisher Scientific | CAT# PA146018 | WB (1:1000) For ChIP |
| Antibody | ERF (Rabbit monoclonal) | Thermo Fisher Scientific | CAT# PA530237 | WB (1:1000) For ChIP |
| Antibody | HSP90 (Rabbit polyclonal) | Cell Signaling | CAT# 4874 S | WB (1:1000) |
| Antibody | Actin (Rabbit monoclonal) | Cell Signaling | CAT# 4970 S | WB (1:1000) |
| Sequence-based reagent | ETV1-CIC-Forward-1 | This paper | ChIP-PCR primers | 5′ CAGGACAAAGAGGAGGCAGCGAGCTG-3′ |
| Sequence-based reagent | ETV1-CIC-Reverse-1 | This paper | ChIP-PCR primers | 5′ GTTTATTGCTGACCCCTCAGCGCTCCGC 3′ |
| Sequence-based reagent | ETV1-ERF-Forward-1- | This paper | ChIP-PCR primers | 5′-CCAGGTCCGGGGTTGAGTGCTGTGC- 3 |
| Sequence-based reagent | ETV1-ERF-Reverse-1 | This paper | ChIP-PCR primers | 5′-CATTTGTGACCAGAACTAGTGACC-3 |
| Sequence-based reagent | ETV1 promoter | SwitchGear Genomics | Product ID: S720645 | |
| Sequence-based reagent | Empty promoter | SwitchGear Genomics | Product ID: S790005 | |
| Chemical compound, drug | BRD32048 | Sigma Aldrich | Cat#: SML1346 | ETV1 inhibitor |

## Cell lines, drug, and reagents

Cell lines were cultured as recommended by the American Type Culture Collection (ATCC) or Sigma. DU-145, PC-3, and HEK293T cells were purchased and authenticated (STR profiling) by ATCC. PNT2 cells were purchased and authenticated (STR profiling) by Sigma. All cell lines were tested for mycoplasma using the e-myco PLUS PCR detection kit (Boca Scientific, Cat#25237). HEK293T cells are human embryonic kidney cells which are commonly used for transfection. PNT2 ERF KD (shERFA1, shERFB1), PNT2 CIC KO (sgCIC1, sgCIC2) and PNT2 ERF KD +CIC KO were derived from parental PNT2 cells. shRNAs targeting ERF to develop PNT2 shERFA1 and PNT2 shERFB1 were obtained from Sigma-Aldrich: TRCN000001391, TRCN0000013912. Puromycin (1 µg/ml) was used as a selection reagent. Two sgRNAs targeting CIC were previously validated and were gifts from William Hahn, Addgene (#74959 and #74953). These sgRNAs were used to develop PNT2 sgCIC1 and PNT2 sgCIC2 cells. Blasticidin (10 µg/ml) was used as a selection agent. PNT2shERFA1 +sgCIC1 and PNT2shERFA1 +sgCIC2 were developed from the combination of the above two shRNA and sgRNAs. All PNT2 cells were grown in DMEM media supplemented with 10% FBS, 100 IU/ml penicillin and 100 µ g/ml streptomycin.

DU-145 ERF, DU-145 CIC KO (sgCIC1, sgCIC2) and DU-145 ERF +CIC KO (ERF +sgCIC1, ERF +sgCIC2) were derived from parental DU-145 cell line. Lentiviral GFP-tagged ERF (GeneCopoeia, EX-S0501-Lv122) was used to develop DU-145 ERF cells with puromycin (1 µg/ml) as a selection marker. The above-mentioned two sgRNAs were used to develop DU-145 sgCIC1 and DU-145 sgCIC2 cells with blasticidin (15 µg/ml) as the selection agent. DU-145 ERF +sgCIC1 and DU-145 ERF +sgCIC2 were developed from the combination of the above two.

PC-3 ERF KD (shERFA1), PC-3 ERF OE, PC-3 CIC OE, PC-3 (ERF +CIC) OE cells were derived from parental PC-3 cell line. shRNAs targeting ERF (Sigma-Aldrich: TRCN000001391) and lentiviral

GFP-tagged ERF (GeneCopoeia, EX-S0501-Lv122) were used to develop PC-3 shERFA1 and PC-3 ERF OE, respectively. PC-3 CIC OE cells were developed using CIC-Myc-tag plasmid purchased from Origene (CAT#: RC215209).

Geneticin (250 µg/ml) was used as a selection agent. PC-3 (ERF +CIC) OE cells were developed using a combination of ERF-GFP and CIC-Myc overexpressing plasmid.

All DU-145 and PC-3 cells were grown in RPMI 1640 media supplemented with 10% FBS, 100 I U/ml penicillin and 100 µ g/ml streptomycin, respectively. All cell lines were maintained at 37 °C in a humidified atmosphere at 5% CO2. All the above mentioned stable cell lines were validated by analyzing the expression of CIC and ERF using qPCR and western blot analysis (*Figure 2—figure supplement 1A*, *Figure 3—figure supplement 1A-C, I*).

BRD32048 is an ETV1 inhibitor that was purchased from Sigma-Aldrich (CAT#:SML1346).

## Analysis of PCa datasets from cBioPortal

15 PCa datasets (see *Figure 1c* for individual studies) were queried for alterations in CIC and ERF using the cBioPortal platform 'query by gene' function. Stratification into 'ERF-CIC No co-deletion' and 'ERF-CIC co-deletion' was performed in cBioPortal and associated with 'Gleason Score', 'AJCC Primary Tumor T Stage', and DFS and PFS using the 'Plots' and 'Comparison/Survival' functions. p-values for comparison between Gleason and Tumor Stage were calculated using FET and survival curves were calculated by Log-rank test.

For the CIC and ERF mutational analysis in primary prostate vs. metastatic castrate resistant PCa, we selected studies that purely represented primary prostate tumors (PNAS 2014, Cell 2014, Nature 2017) and advanced mCRPC (Nature 2012, Cell 2015, PNAS 2019).

## Colony formation assay

Equal number of cells from cell lines (500–600 cells/well) were seeded in a 6-well plate. Cells were allowed to form colonies for 7 days. At day 7, cells were fixed and stained with 0.5% crystal violet solution after washing with PBS and performed in triplicate.

Finally, the colonies with >50 cells were counted under an imageJ software.

## Tumorsphere assay

Approximately 25,000 cells from different groups were cultured in tumorsphere media at 37 °C and 5% CO2 for 7 days. Tumorsphere medium contains serum free DMEM /F12 supplemented with 10 ng/ml FGF (fibroblast growth factor), 20 ng/ml EGF (epidermal growth factor), 1xITS (Insulin-Transferrin-Selenium) and B27 supplement. On day 7, images of different areas of the wells were taken using confocal. The size of the sphere was calculated using Fiji (ImageJ) software in all the tested groups. Each group consisted of three replicate wells and at least 6 images (n>/=6).

## Subcutaneous tumor xenograft assays

Four week old male SCID mice were purchased from Jackson Laboratory. Mice were kept under specific pathogen-free conditions and facilities were approved by the UCSF IACUC. To prepare cell suspensions, PNT2 and its other genetic variants (PNT2 ERF KD, PNT2 CIC KO, PNT2 ERF KD +CIC KO) were briefly trypsinized, quenched with 10% FBS RPMI media and resuspended in PBS. Cells were pelleted again and mixed with PBS/Matrigel matrix (1:1) for a final concentration of $0.1 \times 10^5$ cells/µl. A 100 µl cell suspension containing $1 \times 10^6$ cells were injected (s.c.) in the right and left flanks of immunodeficient mice (n=10/group). Mice were observed for tumor formation in different groups over 7 weeks. For other subcutaneous xenografts, four week old male mice (NU/J) were purchased from Jackson Laboratory and were six-eight weeks old at time of experiment. $1.0 \times 10^6$ DU-145 cells and its variants (DU-145 ERF, DU-145 CICKO and DU-145 ERF +CICKO) were resuspended in PBS/Matrigel (1:1) matrix and injected s.c. into the right and left flanks of nude mice (n=10/group). Tumor volume was measured twice per week using Vernier caliper. Tumor volume was determined using caliper measurements of tumor length (L) and width (W) according to the formula V = (L X W2) X 0.52.

Similarly, $1.0 \times 10^6$ PC-3 tumor cells and its variants (PC-3 ERFOE, CICOE) were injected subcutaneously in flanks of male nude mice (NU/J), n=10/group and tumor volume was monitored in different groups. Mice body weight was measured in all the experiments throughout the study. At the end of the experiment, mice were sacrificed by CO2 overdose in accordance with IACUC guidelines.

## Viability assays

5000 cells were plated in a 12 well plate. Crystal violet staining was performed 5 days after cell plating with 3 replicates per group. CellTiter Glo experiments were performed according to the manufacturer's protocol. In brief, cells were plated in a 96-well plate, and analyzed on a Spectramax microplate reader (Molecular Devices) at different days. Each assay was performed with six replicate wells.

## Transwell invasion assays

RPMI with 10% FBS was added to the bottom well of a trans-well chamber. $2.5 \times 10^4$ cells resuspended in serum free media was then added to the top 8 µm pore matrigel coated (invasion) or non-coated (migration) trans-well insert (BD Biosciences). After 20 hr, non-invading cells on the apical side of inserts were scraped off and the trans-well membrane was fixed in methanol for 15 min and stained with Crystal Violet for 30 min. The basolateral surface of the membrane was visualized with a Zeiss Axioplan II immunofluorescent microscope at 5×. Each trans-well insert was imaged in five distinct regions at 5×and performed in triplicate. % invasion was calculated by dividing the mean # of cells invading through Matrigel membrane / mean # of cells migrating through control insert.

## Wound healing assays

Cells were plated at a density of $0.5 \times 10^6$ cells/well and incubated to form a monolayer in 6-well dishes. Once a uniform monolayer was formed, wound was created by scratching the monolayer with a 1 ml sterile tip. Floating cells were removed by washing the cells with PBS three times. Images of the wound were taken at this point using bright field microscope and considered as a 0 hr time point. Furthermore, media was added in all the wells and cells were left to migrate either for 24 hr (DU-145 cells) or 48 hr (PNT2 ond PC-3 cells). At end point, wound was imaged again using bright field microscope. The wound area at different points was calculated using ImageJ software. Each group consisted of at least three replicate wells.

## DAVID functional clustering analysis

The 91 shared candidate target genes between CIC and ERF identified through our ChIPSeq analysis in PNT2 and VCaP cells, respectively, were used as an input list for analysis using the DAVID Bioinformatics Database (https://david.ncifcrf.gov).

## Real-time quantitative PCR (RT-qPCR)

RT-qPCR was performed in PNT2, DU145 and PC3 cells. Isolation and purification of RNA was performed using RNeasy Mini Kit (Qiagen). 500 ng of total RNA was used in a reverse transcriptase reaction with the SuperScript III firststrand synthesis system (Invitrogen). Quantitative PCR included three replicates per cDNA sample. Human CIC (Cat#. Hs00943425_g1), ERF (Cat#. Hs01100070_g1), ETV1 (Cat#. Hs00951951_m1), ETV4 (Cat#. Hs00383361_g1), ETV5 (Cat#. Hs00927557_m1), and endogenous controls GAPDH (Cat#. Hs02758991_g1) were amplified with Taqman gene expression assay (Applied Biosystems). Expression data were acquired using an ABI Prism 7900HT Sequence Detection System (Thermo Fisher Scientific). Expression of each target was calculated using the 2−ΔΔCt method and expressed as relative mRNA expression.

## Chip-Seq and PCR

ChIP was performed on PNT2 and DU-145 cells with the SimpleChIP Enzymatic Chromatin IP kit, Cell Signaling Technology #9003 in accordance with the manufacturer's protocol. The antibodies used for IP were as follows: CIC (Thermo Fisher Scientific – PA146018) and ERF (Thermo Fisher Scientific –PA530237). Paired-end 150 bp (PE150) sequencing on an Illumina HiSeq platform was then performed. ChIP-Seq peak calls were identified through Mode-based Analysis of ChIP-Seq (MACS). For ETV1 ChIP-PCR validation, primers were designed in the proximal regulatory element of ETV1. The promoter primer sequences are listed in supplementary experimental methods:

VCaP ERF ChIP-Seq was previously performed (*Bose et al., 2017*) and publicly available in the GEO database: GSE83653. For this analysis, the samples used are listed in supplementary experimental methods.

Promoter primer sequences for ChIP-PCR and ChIP-Seq are provided in the Supplementary Methods (*Supplementary file 2*).

### GSM2612455_INPUT red replicate two luciferase promoter assay

293T and DU-145 cells were split into a 96 well plate to achieve 50–70% confluence the day of transfection. LightSwitch luciferase assay system (SwitchGear Genomics) was used per the manufacturer's protocol. Briefly, a mixture containing FuGENE 6 transfection reagent, 50 ng Luciferase GoClone ETV1 promoter (S720645) plasmid DNA, 50 ng of either control (empty) vector or fully sequenced ERF cDNA (GeneCopoeia [EX-S0501Lv122]), were added to each well. All transfections were performed in quintuplicate. The plates were assessed for luciferase activity after 48 hr of treatment.

### Single-sample gene set enrichment analysis (ssGSEA)

Gene-level expression of our dual CIC and ERF deficient PNT2 cells are computed using RSEM (*Setlur et al., 2008*) (Version 1.3.3) and log-2 normalized. The signature gene set of dual *CIC* and *ERF* deficient PNT2 cells (*ERF* KD +*CIC* KO) is defined as the top upregulated genes compared to the CIC and ERF WT cells. The ssGSEA module on GenePattern was then used to project the TCGA-PRAD dataset onto the transcriptional space defined by both the *ERF* KD +*CIC* KO signature gene set and previously established gene sets including the *ETV1*-regulated gene set, *ERF* KD signature and *TMPRSS2-ERG* fusion signature. The ssGSEA enrichment scores of the *ERF* KD +*CIC* KO signature gene set for the TCGA-PRAD samples are compared with the scores of the other signature gene sets and visualized in heatmaps and used for downstream association analyses.

### Gene knockdown (KD) and OE assays

ON-TARGET plus Scramble, ETV1 (L-003801-00-0005) and CIC (L-015185-01-0005) siRNAs were obtained from GE Dharmacon and transfection was performed with Dharmafect transfection reagent per manufacturer recommendations. ETV1 inhibitorBRD32048 (SML1346) was purchased from Millipore Sigma. Lentiviral GFP-tagged ERF was obtained from GeneCopoeia (EX-S0501-Lv122). pCMV-CIC with myc-tag was purchased from Origene and validated previously.

### Experimental plan for experiments using ETV1 inhibitor

Cells were plated at a density of $0.2\times10^6$ in a 6-well plate. Next day, cells were transfected with siScramble and siCIC. After 48 hr, 25 µM and 50 µM of BMS32048 (ETV1 inhibitor) was added to both the transfected groups and control groups were treated with DMSO. 24 hr post ETV1 inhibitor treatment, cells from all the groups were replated for viability assay in a 24-well plate (n=3) and transwell invasion assay (n=3). Viability assay was performed using crystal violet. The relative density of crystal violet and % invasion was calculated comparing siCIC with siSCRM conditions in each respective drug concentration.

### Western blot analysis

Adherent cells were washed and lysed with RIPA buffer supplemented with proteinase and phosphatase inhibitors. Proteins were separated by SDS-PAGE, transferred to nitrocellulose membranes and blotted with antibodies recognizing: CIC (Thermo Fisher Scientific –PA146018), ERF (Thermo Fisher Scientific –PA530237), ETV1 (Thermo Fisher Scientific –MA515461), HSP90 (Cell Signaling– 4874 S), Actin (Cell Signaling – 4970 S). All immunoblots represent at least two independent experiments.

### Co-immunoprecipitation assays

293T cells were transfected with GFP-tagged ERF for 48 hr, lysed, quantified, and incubated with either IgG (Cell Signaling Technology; 2729) fused to Dynabeads Protein G (Thermo Fisher Scientific; 10004D) or anti-GFP magnetic beads overnight at 4 °C. Proteins were separated by SDS-PAGE, transferred to nitrocellulose membranes, and blotted with antibodies recognizing GFP or CIC. Myc-tagged CIC was transfected into HEK293T cells for 48 hr, lysed, quantified, and incubated with either IgG (Cell Signaling Technology; 2729) fused to Dynabeads Protein G (Thermo Fisher Scientific; 10004D) or anti-myc magnetic beads overnight at 4 °C. Proteins were separated by SDS-PAGE, transferred to nitrocellulose membranes, and blotted with antibodies recognizing Myc or ERF.

### Statistical analysis

Experimental data are presented as mean +/-Standard Deviation (SD) or Standard Error of the Mean (SEM). P-values derived for all in-vitro experiments were calculated using two-tailed student's t-test

or one-way ANOVA. The detailed statistical analysis performed for each experiment is defined in the figure legends.

## Study approval

For tumor xenograft studies, specific pathogen-free conditions and facilities were approved by the American Association for Accreditation of Laboratory Animal Care. Surgical procedures were reviewed and approved by the UCSF Institutional Animal Care and Use Committee (IACUC), protocol #AN178670-03.

## Acknowledgements

The authors thank members of the Okimoto and Huang labs and funding support from the following sources: A National Cancer Institute (NCI) Cancer Center Support Grant (P30CA082103) and Benioff Initiative for Prostate Cancer Research Awards (N.G and R.A.O), the Basic Science Research Program through the National Research Foundation of Korea (NRF) funded by the Ministry of Education (NRF- 2020R1A6A3A03039483)(J.W.K) and NCI K08-CA222625 and R37CA255453 awards to R.A.O.

## Additional information

### Funding

| Funder | Grant reference number | Author |
| --- | --- | --- |
| National Cancer Institute | K08CA222625 | Ross A Okimoto |
| National Cancer Institute | 5R37CA255453 | Ross A Okimoto |
| National Cancer Institute | P30CA082103 | Ross A Okimoto |
| Benioff Initiative for Prostate Cancer Research Awards | | Felix Y Feng |
| National Research Foundation of Korea | NRF-2020R1A6A3A03039483 | Ji Won Kim |

The funders had no role in study design, data collection and interpretation, or the decision to submit the work for publication.

### Author contributions

Nehal Gupta, Conceptualization, Data curation, Formal analysis, Investigation, Writing – original draft, Writing – review and editing; Hanbing Song, Data curation, Formal analysis, Writing – review and editing; Wei Wu, Yone K Lin, Formal analysis, Writing – review and editing; Rovingaile K Ponce, Ji Won Kim, Data curation, Writing – review and editing; Eric J Small, Supervision, Writing – original draft, Writing – review and editing; Felix Y Feng, Supervision, Funding acquisition, Writing – review and editing; Franklin W Huang, Data curation, Supervision, Writing – original draft, Writing – review and editing; Ross A Okimoto, Conceptualization, Formal analysis, Supervision, Funding acquisition, Writing – original draft, Writing – review and editing

### Author ORCIDs

Nehal Gupta ![ORCID] http://orcid.org/0000-0002-8931-5759
Franklin W Huang ![ORCID] http://orcid.org/0000-0001-5447-0436
Ross A Okimoto ![ORCID] http://orcid.org/0000-0002-4467-8476

### Ethics

For tumor xenograft studies, specific pathogen-free conditions and facilities were approved by the American Association for Accreditation of Laboratory Animal Care. Surgical procedures were reviewed and approved by the UCSF Institutional Animal Care and Use Committee (IACUC), protocol #AN178670-03.

Decision letter and Author response
Decision letter https://doi.org/10.7554/eLife.77072.sa1
Author response https://doi.org/10.7554/eLife.77072.sa2

## Additional files

### Supplementary files
- Supplementary file 1. List of 91 CIC and ERF shared putative target genes used for DAVID analysis.
- Supplementary file 2. Promoter primer sequences for ChIP-PCR and ChIP-Seq experiments.
- Transparent reporting form

### Data availability
All sequencing data including RNASeq and ChIP seq have been deposited in GEO under accession codes GSE216732 and GSE216318, respectively.

The following datasets were generated:

| Author(s) | Year | Dataset title | Dataset URL | Database and Identifier |
|---|---|---|---|---|
| Gupta N, Song H, Wu W, Kriska Ponce R, Lin Y, Kim JW, Small EJ, Feng FY, Huang FW, Okimoto RA | 2022 | Decoding the Protein Composition of Whole Nucleosomes with Nuc-MS | https://www.ncbi.nlm.nih.gov/geo/query/acc.cgi?acc=GSE216732 | NCBI Gene Expression Omnibus, GSE216732 |
| Gupta N, Song H, Wu W, Kriska Ponce R, Lin Y, Kim JW, Small EJ, Feng FY, Huang FW, Okimoto RA | 2022 | The CIC-ERF co-deletion underlies fusion independent activation of ETS family member, ETV1, to drive prostate cancer progression | https://www.ncbi.nlm.nih.gov/geo/query/acc.cgi?acc=GSE216318 | NCBI Gene Expression Omnibus, GSE216318 |

The following previously published dataset was used:

| Author(s) | Year | Dataset title | Dataset URL | Database and Identifier |
|---|---|---|---|---|
| Bose R | 2017 | Loss of Function Mutations in ETS2 Repressor Factor (ERF) Reveal a Balance Between Positive and Negative ETS Factors Controlling Prostate Oncogenesis | https://www.ncbi.nlm.nih.gov/geo/query/acc.cgi?acc=GSE83653 | NCBI Gene Expression Omnibus, GSE83653 |

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
