## [Editor Report]

This study provides insight into a potentially new genetically defined subset of prostate tumors driven by concurrent loss of the ERF and CIC tumor suppressor genes, in the absence of the canonical fusion event involving TMPRSS2 (around 10% of all cases). The work both validates previous findings and provides new data that support a compelling overall conclusion that combined ERF and CIC loss promotes prostate tumorigenesis by increasing expression of the oncogenic driver ETV1. This is an important study based on convincing evidence, that will be of interest to researchers in the field of prostate cancer.

---

## [Decision Letter]

**Decision letter after peer review:**

Thank you for submitting your article "The CIC-ERF co-deletion underlies fusion independent activation of ETS family member, ETV1, to drive prostate cancer progression" for consideration by *eLife*. Your article has been reviewed by 3 peer reviewers, and the evaluation has been overseen by a Reviewing Editor and Erica Golemis as the Senior Editor. The following individual involved in the review of your submission has agreed to reveal their identity: Stephen Yip (Reviewer #3).

Essential revisions:

1. Can the authors confirm that the 15 cBioPortal datasets are really independent? To clarify this, it would be informative to have the numbers presented for each dataset in the associated Figure 1c legend. For example, the reader should then be able to see that the TCGA 2018 and TCGA numbers are quite small as most of these cases are also in TCGA 2015. Figure 1c is also hard to interpret given the data has been sorted (we assume) by CIC and ERF alterations. It would be more informative for the reader if it was sorted according to the study so we can see the distribution of alterations across each study. This then ties in with the later analysis looking at the difference in alterations between primary and metastatic prostate cancer datasets. It is also hard to interpret the stage data presented in Figure 1d, is it possible to condense the stages somehow? For example, condense T2A, T2B and T2C into one category (T2), etc.

2. Can the numbers of cases present in the primary and metastatic prostate cancer groups be specified in the text or in Figure 1e. Similarly, Can the number of cases present in each group of the survival analyses be specified in the text or in Figure 1g? Without any of the numbers requested above, it is difficult to determine the statistical power these analyses have.

3. The sample size used in each of the cell line assays should also be presented in the corresponding figure legends as (n=x). This includes bioreps of the cell culture experiments.

4. Supplementary Table 1 is missing.

5. The description of the ssGSEA work is very unclear both in the methods and Results sections. GSEA also stands for Gene Set Enrichment Analysis, not Gene Expression Analysis as stated in the results. It is unclear whether the authors used gene expression data from PNT2 cells or if they used data from the TCGA-PRAD dataset. If they used PNT2 gene expression data, how was this generated and how exactly was the TCGA-PRAD dataset involved? A more comprehensive and coherent description of these analyses is required in both the methods and Results sections. This includes the figure legend (4m) where there is no explanation for the abbreviations used.

6. The figures associated with the results describing the assessment of the ETV1 inhibitor in DU-145 cells actually show LNCap cells (Figure 4A-c). Please clarify.

7. The authors say they observe enhanced sensitivity to BRD32048 in DU-145 CIC KO cells, but this is not significant and should be noted. Why wasn't this work repeated in the other cell lines (PNT2 and PC-3, especially, given their contexts in prostate cancer metastatic potential) or in vivo? Or why not reconstitute ERF in DU-145 to see the effect of the inhibitor on CIC KO/KD alone? These experiments are essential to strengthen and generalize conclusions.

8. The supplementary figure (4d-f) also shows results from PC-3 cells, but this is not mentioned in the text.

9. The sub-figures (5b-5d) are in the wrong order, 5b shows invasiveness while 5c shows viability.

10. The methods section provides no information as to how the authors conducted their cell culture experiments using the ETV1 inhibitor (including technical/bioreps etc.), and as such, are not reproducible. Overall, the methods section is very disjointed and the methodology should be presented in the order that it is described in the results (aside from the cell line/culture information). This would also allow the authors to get rid of some repetition that is currently present and ensure there is still no missing methodology. In addition to some of the more specific methodology suggestions mentioned previously, can more information regarding the DAVID functional clustering analysis be added, including a reference/web link if available?

11. The authors should perform additional replicates to provide statistical significance for experiments currently lacking significance or temper their claims. For example, Figure 4l should probably be removed, unless additional replicates improve the data.

12. If ERF and CIC co-deletion function by increasing ETV1 transcription, then an increase in ETV1 mRNA levels in these tumors should be apparent in patient tumor datasets, such as those analyzed in Figure 1. This is a key missing analysis. This needs to be supplied.

13. A growth curve to measure proliferation rate changes after CIC and ERF manipulation would provide more rigor. Does ERF overexpression inhibit phenotypes in PNT2 cells?

14. Testing a few negative control genomic regions and showing no enrichment would alleviate the concern regarding controls for ChIP-PCR.

15. This phrase in the introduction needs references: "…and inactivation of CIC de-represses ETV1, ETV4, and ETV5 transcription to promote tumor progression."

16. It is important to investigate potential proteomic interactions between CIC and ERF using Western blot and immunoprecipitation. Whether they bind to one another and what happens when only CIC or ERF is not expressed. What happens to the level of CIC, or ERF, in the absence of the other protein and the consequence of this, if there is altered expression.

[Editors’ note: further revisions were suggested prior to acceptance, as described below.]

Thank you for resubmitting your work entitled "The CIC-ERF co-deletion underlies fusion independent activation of ETS family member, ETV1, to drive prostate cancer progression" for further consideration by *eLife*. Your revised article has been evaluated by Erica Golemis (Senior Editor) and a Reviewing Editor.

The manuscript has been improved but there are some remaining issues that need to be addressed, as outlined below:

In general, no further experiments are needed, and the remaining critiques can be addressed by writing changes. Reviewer 3 is satisfied with the current version of the article, and Reviewer 1 has points that are largely directed at clarity; please address these. However, Reviewer 2 has residual issues with the significance and interpretation of the data. Please address point 1 in the discussion; for point 2, we suggest you present alternative analyses of the data (perhaps in a new supplemental figure) that allow the comparisons suggested, and discuss the significance of the results obtained through this analysis.

*Reviewer #1 (Recommendations for the authors):*

The authors have addressed the majority of my concerns, however, I don't feel I can comment on whether they have adequately addressed the other reviewers' comments. I have just a few points below that require the authors' attention.

1. As suggested, the authors have included a Table (Supplementary Table 1) that presents the CIC-ERF data for the various studies. However, I'm not sure the authors understood our comment regarding study overlap. For example, they present two rows of data for TCGA samples (PanCancer Atlas and Firehose Legacy) but these studies essentially consist of the same samples (the second just has a few more samples included). Thus, it is somewhat misleading to present two primary predominant cohorts with lower CIC-ERF alterations, when this is really just a single cohort. The authors also need to determine how many samples overlap the two SU2C/PCF studies and the MSK studies. This also applies to the data presented in Supplementary Figure 1 and impacts the statement in the Results section that says "We analyzed over 6047 PCa tumors from 5839 patients" (line 99), as this is not the case given numbers overlap across the studies.

Figure 1C still appears to be sorted by something other than study. If it was sorted by study, the top bar in this figure would have, for example, all the MSK-IMPACT prostate samples grouped together, all the PRAD (MSKCC, 2020) samples grouped together, etc. So there would be a solid teal block, a solid pink block, etc. At the moment you can see that on the far left of the "Study" bar, it starts with a mix of green and purple study samples, and so on.

2. The first and second sentences of the abstract now essentially say the same thing. The second sentence can be removed.

3. Ensure all gene names are italicised.

4. There is an issue with one of the references (TCGA 2015 publication).

*Reviewer #2 (Recommendations for the authors):*

Overall the revision has not alleviated my concerns regarding the viability of the overall thesis that ERG and CIC co-deletion leading to activation of ETV1 is a driver of prostate cancer progression. There are two main reasons for this. First is the finding that there is no statistically significant difference in ETV1 expression between patient tumors with wild-type ERF and CIC, and co-deleted ERF and CIC. This seems to be a major problem. Second much of the critical data is underwhelming, does not reach statistical significance, and seems to be presented in ways intended to obscure the real results. Examples of this include the setting of every odd column to 1 and normalization of each even column to the prior odd column in Figures 5A, 5B, 5C, and 5S2B, C, E, and F – with no explanation that this is being done. This is also true of the ChIP control presented in the rebuttal, which includes an irrelevant input column that obscures the necessary comparison of IgG to anti-CIC and anti-ERF. It is also a problem that this control is presented in an entirely different way than the ChIP in Figure 4 that it is necessary to control for, and the control does not appear to be included in the revised manuscript.

*Reviewer #3 (Recommendations for the authors):*

I am satisfied with the authors' responses and the additional experiments performed. I am somewhat disappointed that the selective knockdown of CIC and ERF did not further elucidate the functional interactions between these two proteins but I appreciate the effort and the underlying complexity of the interactions which could also be affected by cell and developmental contexts.

---

## [Author Response]

Essential revisions:1. Can the authors confirm that the 15 cBioPortal datasets are really independent? To clarify this, it would be informative to have the numbers presented for each dataset in the associated Figure 1c legend. For example, the reader should then be able to see that the TCGA 2018 and TCGA numbers are quite small as most of these cases are also in TCGA 2015. Figure 1c is also hard to interpret given the data has been sorted (we assume) by CIC and ERF alterations. It would be more informative for the reader if it was sorted according to the study so we can see the distribution of alterations across each study. This then ties in with the later analysis looking at the difference in alterations between primary and metastatic prostate cancer datasets. It is also hard to interpret the stage data presented in Figure 1d, is it possible to condense the stages somehow? For example, condense T2A, T2B and T2C into one category (T2), etc.

We agree with the reviewer and provide a detailed table with (1) the total number of cases in each study; (2) the type of CIC and ERF deletion (shallow or deep); and (3) the percentage of cases in each study with a CIC-ERF co-deletion. This analysis further suggests that the CIC-ERF co-deletion is enriched in studies that were predominantly composed of advanced prostate cancers (highlighted in yellow) compared to studies predominantly composed of primary prostate cancer (highlighted in green). We have included this table as Figure 1 – Table Supplement 1.

In order to illustrate the frequency of CIC-ERF co-deletions in each study we sorted according to the study as recommended by the reviewer. These data indicate that CIC-ERF co-deletions are observed at a higher frequency in studies that predominantly analyzed advanced prostate cancer relative to primary prostate cancers. We have incorporated this figure into our revised manuscript as Figure 1 – Supplementary Figure 1.

With respect to the staging data, we utilized the presorted selection in cBioPortal and were unfortunately, unable to further condense the sub-stages into a single staging category (ie. T2A, T2B, T2C into T2).

2. Can the numbers of cases present in the primary and metastatic prostate cancer groups be specified in the text or in Figure 1e.

We agree with the reviewer and have now incorporated detailed information on the number of cases in each of the primary and metastatic prostate cancer groups. We provide this information below and have incorporated these numbers into the text.

Primary prostate cancer cohorts:

PNAS 2014. Hieronymus et al.

272 primary prostate cancers

Cell 2015. TCGA

333 primary prostate cancers

Nature 2017. Fraser et al.

477 primary (localized, non-indolent) prostate cancer.

Metastatic prostate cancer cohorts:

Nature 2012. Grasso et al.

50 metastatic CRPCs and 11 treatment-naive, high-grade localized prostate cancers.

Cell 2015. Robinson et al.

150 mCRPC

PNAS 2019. Abida et al.

429 mCRPC

Similarly, Can the number of cases present in each group of the survival analyses be specified in the text or in Figure 1g? Without any of the numbers requested above, it is difficult to determine the statistical power these analyses have.

Again, we agree that adding these numbers are informative and have incorporated them into the text.

Disease-free survival (DFS):

Altered: 90 (# of total) 25 (# of events)

Non-altered 910 (# of total) 153 (# of events)

Progression-free survival (PFS):

Altered: 52 (# of total) 16 (# of events)

Non-altered: 442 (# of total) 77(# of events)

3. The sample size used in each of the cell line assays should also be presented in the corresponding figure legends as (n=x). This includes bioreps of the cell culture experiments.

We have now incorporated the sample size for each experiment in the corresponding figure legends.

4. Supplementary Table 1 is missing.

We apologize for this oversight and have now incorporated this table as Supplemental Table 2 in the revised version of our manuscript.

5. The description of the ssGSEA work is very unclear both in the methods and Results sections. GSEA also stands for Gene Set Enrichment Analysis, not Gene Expression Analysis as stated in the results. It is unclear whether the authors used gene expression data from PNT2 cells or if they used data from the TCGA-PRAD dataset. If they used PNT2 gene expression data, how was this generated and how exactly was the TCGA-PRAD dataset involved? A more comprehensive and coherent description of these analyses is required in both the methods and Results sections. This includes the figure legend (4m) where there is no explanation for the abbreviations used.

We agree with the Reviewer and have expanded the methods and Results sections to include additional details on how we performed the ssGSEA. We also apologize for the oversight, we have changed Gene Expression Analysis to Gene Set Enrichment Analysis.

6. The figures associated with the results describing the assessment of the ETV1 inhibitor in DU-145 cells actually show LNCap cells (Figure 4A-c). Please clarify.

We have added the explanation describing LNCaP cells for supplementary figure (4a-c)

7. The authors say they observe enhanced sensitivity to BRD32048 in DU-145 CIC KO cells, but this is not significant and should be noted. Why wasn't this work repeated in the other cell lines (PNT2 and PC-3, especially, given their contexts in prostate cancer metastatic potential) or in vivo? Or why not reconstitute ERF in DU-145 to see the effect of the inhibitor on CIC KO/KD alone? These experiments are essential to strengthen and generalize conclusions.

We thank the reviewer for this comment. We have now performed additional studies to address this concern. In particular, we genetically silenced *CIC* and *ERF* in PNT2 cells and observed enhanced sensitivity to the ETV1 inhibitor. Consistent with these results we also knocked down *ERF* in *CIC* deficient PC-3 cells and again noted an increase in ETV1 inhibitor sensitivity compared to scramble control. These data are incorporated into the revised manuscript as Figure 5 —figure supplement 1A-F.

Finally, we reconstituted ERF into ERF deficient DU145 cells with CIC KO. We did not observe a significant difference in ETV1 inhibitor sensitivity in this setting.

8. The supplementary figure (4d-f) also shows results from PC-3 cells, but this is not mentioned in the text.

We apologize for this oversight and have now mentioned PC-3 cells in the text.

9. The sub-figures (5b-5d) are in the wrong order, 5b shows invasiveness while 5c shows viability.

We apologize for this error. We have corrected the order of the sub-figure.

10. The methods section provides no information as to how the authors conducted their cell culture experiments using the ETV1 inhibitor (including technical/bioreps etc.), and as such, are not reproducible. Overall, the methods section is very disjointed and the methodology should be presented in the order that it is described in the results (aside from the cell line/culture information). This would also allow the authors to get rid of some repetition that is currently present and ensure there is still no missing methodology. In addition to some of the more specific methodology suggestions mentioned previously, can more information regarding the DAVID functional clustering analysis be added, including a reference/web link if available?

We have added a paragraph in the methods section describing the methodology using the ETV1 inhibitor. Overall, we have also restructured the methods section and the methodology is presented in the order as they appear in the Results section. We have also included a web link and references to DAVID in the manuscript text and methods sections.

11. The authors should perform additional replicates to provide statistical significance for experiments currently lacking significance or temper their claims. For example, Figure 4l should probably be removed, unless additional replicates improve the data.

We completely agree with the reviewer. We have removed figure 4l from the manuscript.

12. If ERF and CIC co-deletion function by increasing ETV1 transcription, then an increase in ETV1 mRNA levels in these tumors should be apparent in patient tumor datasets, such as those analyzed in Figure 1. This is a key missing analysis. This needs to be supplied.

We used the largest prostate cancer dataset (n=455) and observed a trend but not a statistically significant difference in ETV1 expression in tumors that harbored the CIC-ERF co-deletion compared to CIC-ERF replete tumors.

**Author response image 1. sa2fig1:** 

13. A growth curve to measure proliferation rate changes after CIC and ERF manipulation would provide more rigor. Does ERF overexpression inhibit phenotypes in PNT2 cells?

We assessed the growth rate (over 7 days) of PNT2 cells engineered to express *shERF*, *sgCIC* or combinatorial silencing of ERF (*shERF*) and CIC (*sgCIC*). We observed an increase in growth rate in PNT2 cells with dual ERF and CIC suppression compared to PNT2 control and either *shERF* or *sgCIC* alone.

We overexpressed ERF in PNT2 cells but did not observe a significant difference in viability (crystal violet or cell-titer glo assays) or in colony formation.

**Author response image 3. sa2fig3:** 

14. Testing a few negative control genomic regions and showing no enrichment would alleviate the concern regarding controls for ChIP-PCR.

We used two commercially validated human negative control q-pcr primer sets from Active Motif as an additional layer of control to address this concern. Using these primers targeting Gene Deserts on chromosome 12 and 4, we made comparisons to either IgG control or Input (2%).

**Author response image 4. sa2fig4:** 

15. This phrase in the introduction needs references: "…and inactivation of CIC de-represses ETV1, ETV4, and ETV5 transcription to promote tumor progression."

We have added the references for the indicated phrase.

16. It is important to investigate potential proteomic interactions between CIC and ERF using Western blot and immunoprecipitation. Whether they bind to one another and what happens when only CIC or ERF is not expressed. What happens to the level of CIC, or ERF, in the absence of the other protein and the consequence of this, if there is altered expression.

We agree with the reviewer and have performed co-immunoprecipitation experiments to address these concerns. Specifically, using HEK293T cells we expressed and pulled down GFP-tagged ERF, probing for CIC by western blot. CIC did not co-immunoprecipitate with GFP-tagged ERF under these conditions. The reciprocal experiment was also performed through expression of Myc-tagged CIC in HEK293T cells. ERF did not co-immunoprecipitate with CIC under these conditions. These data indicate that CIC and ERF do not interact in HEK293T cells. These studies have been incorporated to the revised manuscript as Figure 4 —figure supplement 1K-L.

We next performed ERF knockdown (KD) and ERF overexpression (OE) studies in HEK293T cells as requested by the Reviewer. Interestingly, we observed an increase in CIC expression following either ERF KD or ERF OE.

**Author response image 5. sa2fig5:** 

Moreover, we assessed ERF expression following CIC knockout or CIC overexpression in 293T cells. CIC expression increased using a single crispr-based sgRNA (sgCIC1) but this finding was not consistently observed with sgCIC2. CIC overexpression also increased ERF expression compared to control. Collectively, these findings indicate that the functional relationship between CIC and ERF may be complex and warrants further investigation into how these two proteins regulate one another. This will be an area of future interest and investigation.

**Author response image 6. sa2fig6:** 

[Editors’ note: further revisions were suggested prior to acceptance, as described below.]

Reviewer #1 (Recommendations for the authors):The authors have addressed the majority of my concerns, however, I don't feel I can comment on whether they have adequately addressed the other reviewers' comments. I have just a few points below that require the authors' attention.1. As suggested, the authors have included a Table (Supplementary Table 1) that presents the CIC-ERF data for the various studies. However, I'm not sure the authors understood our comment regarding study overlap. For example, they present two rows of data for TCGA samples (PanCancer Atlas and Firehose Legacy) but these studies essentially consist of the same samples (the second just has a few more samples included). Thus, it is somewhat misleading to present two primary predominant cohorts with lower CIC-ERF alterations, when this is really just a single cohort. The authors also need to determine how many samples overlap the two SU2C/PCF studies and the MSK studies. This also applies to the data presented in Supplementary Figure 1 and impacts the statement in the Results section that says "We analyzed over 6047 PCa tumors from 5839 patients" (line 99), as this is not the case given numbers overlap across the studies.

We agree with the reviewer. We have removed the statement “we analyzed over 6047 PCa tumors from 5839 patients”.

We also incorporated the following statement within the text to hopefully bring additional clarity. “One limitation of this analysis is that individual studies included overlapping tumors from the same patient (ie. TCGA PanCancer and Firehose Legacy cohorts).” Moreover, we emphasized in the text that we used this correlative analysis to generate a hypothesis to explore the functional interaction between CIC and ERF in the context of prostate cancer progression.

Figure 1C still appears to be sorted by something other than study. If it was sorted by study, the top bar in this figure would have, for example, all the MSK-IMPACT prostate samples grouped together, all the PRAD (MSKCC, 2020) samples grouped together, etc. So there would be a solid teal block, a solid pink block, etc. At the moment you can see that on the far left of the "Study" bar, it starts with a mix of green and purple study samples, and so on.

Figure 1C was sorted by gene alteration. For this analysis, we sorted by *CIC* and aligned this with *ERF* alterations. In response to the Reviewer, we previously incorporated the frequency of CIC and ERF co-deletions found in each cohort/study in Figure 1 – Supplementary Figure 1.

2. The first and second sentences of the abstract now essentially say the same thing. The second sentence can be removed.

We agree with the Reviewer and have removed the first sentence of the Abstract.

3. Ensure all gene names are italicised.

We have reviewed the manuscript and italicized all gene names.

4. There is an issue with one of the references (TCGA 2015 publication).

We thank the reviewer for identifying this issue. We have corrected the reference.

Reviewer #2 (Recommendations for the authors):Overall the revision has not alleviated my concerns regarding the viability of the overall thesis that ERG and CIC co-deletion leading to activation of ETV1 is a driver of prostate cancer progression. There are two main reasons for this. First is the finding that there is no statistically significant difference in ETV1 expression between patient tumors with wild-type ERF and CIC, and co-deleted ERF and CIC. This seems to be a major problem.

We have expanded the Discussion section to include a paragraph on these findings.

“A limitation of this targeted approach in prostate cancer patients is that we did not find a statistically significant difference in *ETV1* transcript levels in *CIC-ERF* co-deleted tumors compared to *CIC-ERF* replete tumors in the TCGA-PRAD (n=455) dataset. One potential explanation is that ETV1 is commonly upregulated in human prostate cancers through mechanisms beyond CIC-ERF loss. Thus, while our data support the upregulation and potential induced dependence on ETV1 in our *CIC-ERF* deficient systems, it remains unclear if this will translate beyond our cell-line based models into prostate cancer patients that harbor *CIC-ERF* co-deletions. Thus, larger studies that aim to evaluate: (1) ETV1 mRNA and protein expression levels; (2) the biological significance of ETV1 function; and (3) the clinical application of ETV1 inhibitors in patients with endogenous *CIC-ERF* co-deleted tumors (compared to *CIC-ERF* WT tumors) is warranted in this subset of PCa. Collectively, we have uncovered a molecular subset of PCa defined by a co-deletion of CIC and ERF and further demonstrate a mechanismbased strategy to potentially limit tumor progression through ETV1 inhibition in this subset of human PCa.”

Following ERG, ETV1 is the most frequently overexpressed ETS gene in prostate cancers. Prior studies (PMIDs: 16254181, 18594527, 21298110) have found both fusion dependent (*ETV1*-translocations) and fusion independent (increased WT *ETV1* transcript levels) mechanisms that can lead to increased *ETV1* mRNA expression in prostate cancers. Thus, increased *ETV1* expression can be observed in other subsets of prostate cancer beyond *CIC-ERF* co-deleted tumors. We suspect that our analysis comparing *CIC-ERF* co-deletion to *CIC-ERF* WT tumors did not reach statistical significance due to the inability to identify other prostate tumors that have increased *ETV1* expression from alternative mechanisms, beyond CIC-ERF co-deletions.

Second much of the critical data is underwhelming, does not reach statistical significance, and seems to be presented in ways intended to obscure the real results. Examples of this include the setting of every odd column to 1 and normalization of each even column to the prior odd column in Figures 5A, 5B, 5C, and 5S2B, C, E, and F – with no explanation that this is being done.

We have now modified the Figure title and clearly delineated the comparisons in the figure 5 legend. It now reads:

Figure 5. Combinatorial CIC and ERF loss can modulate ETV1 inhibitor sensitivity in prostate cells. (A) DU-145 cells were transfected with either siScramble (siSCM) or siCIC. After 48 hours, BRD32048 (ETV1 inhibitor) was added to both the transfected groups at the defined concentrations (0 μM, 25 μM, 50 μM). After 24 hours of BRD32048 treatment, cells were replated for crystal violet assay (0.4%) and images were taken and analyzed after 5 days (n=3). siCIC was compared to siSCRM conditions in each respective drug concentration. (B) Crystal violet viability assay (n=3). siETV1 groups were compared to siSCRM control groups +/- CIC expression (sgCtrl, sgCIC1, or sgCIC2). (C) DU-145 cells were transfected with either siScramble (siSCM) or siCIC. After 48 hours, BRD32048 was added to the transfected groups at defined concentrations (0 μM or 50 μM). Transwell invasion assays (n=3) were performed 24 hours after the addition of BRD32048. siCIC was compared to siSCRM in the 0 μM and 50 μM concentration groups. (D) Transwell invasion assays (n=3) comparing siETV1 to siSCRM control +/- CIC expression (sgCtrl, sgCIC1 or sgCIC2). P value = *p<0.05, **p<0.01 for all figures. Error bars represent SD.

Moreover, we now provide additional figures (Figure 5 —figure supplement 2D,E,I,J) to allow for comparisons between the PNT2 and PC-3 crystal violet assays in Figure 5 —figure supplement 2B, C, G, and H. We continue to observe a decrease in invasiveness and a milder, albeit consistent decrease in viability upon ETV1 inhibition in our engineered CIC-ERF deficient cells. We have further modified the text to highlight the shortcomings of our in vitro models and reserve claims on the potential impact on clinical translation until further studies are performed as we mention in the Discussion section.

This is also true of the ChIP control presented in the rebuttal, which includes an irrelevant input column that obscures the necessary comparison of IgG to anti-CIC and anti-ERF. It is also a problem that this control is presented in an entirely different way than the ChIP in Figure 4 that it is necessary to control for, and the control does not appear to be included in the revised manuscript.

We have included the ChIP control experiments into the revised manuscript as Figure 4

—figure supplement 1O-P. In addition, we quantified the *ETV1* ChIP-PCR performed in Figure 4 using conventional densitometry (Image J), making comparison between CIC or ERF to IgG. This has been included in the revised manuscript as Figure 4 —figure supplement 1M-N.